# An Overview of Several Inhibitors for Alzheimer’s Disease: Characterization and Failure

**DOI:** 10.3390/ijms221910798

**Published:** 2021-10-06

**Authors:** Subramanian Boopathi, Adolfo B. Poma, Ramón Garduño-Juárez

**Affiliations:** 1Instituto de Ciencias Físicas, Universidad Nacional Autónoma de México, Cuernavaca 62210, Mexico; boopathi@icf.unam.mx; 2Department of Biosystems and Soft Matter, Institute of Fundamental Technological Research Polish Academy of Science, Pawińskiego 5B, 02-106 Warsaw, Poland; 3International Center for Research on Innovative Biobased Materials (ICRI-BioM)—International Research Agenda, Lodz University of Technology, Zeromskiego 116, 90-924 Lodz, Poland; adolfo.poma-bernaola@p.lodz.pl

**Keywords:** Alzheimer’s disease, amyloid β peptide, plaque formation, small molecules, M30, gabapentin, MD simulation

## Abstract

Amyloid beta (Aβ) oligomers are the most neurotoxic aggregates causing neuronal death and cognitive damage. A detailed elucidation of the aggregation pathways from oligomers to fibril formation is crucial to develop therapeutic strategies for Alzheimer’s disease (AD). Although experimental techniques rely on the measure of time- and space-average properties, they face severe difficulties in the investigation of Aβ peptide aggregation due to their intrinsically disorder character. Computer simulation is a tool that allows tracing the molecular motion of molecules; hence it complements Aβ experiments, as it allows to explore the binding mechanism between metal ions and Aβ oligomers close to the cellular membrane at the atomic resolution. In this context, integrated studies of experiments and computer simulations can assist in mapping the complete pathways of aggregation and toxicity of Aβ peptides. Aβ oligomers are disordered proteins, and due to a rapid exploration of their intrinsic conformational space in real-time, they are challenging therapeutic targets. Therefore, no good drug candidate could have been identified for clinical use. Our previous investigations identified two small molecules, M30 (2-Octahydroisoquinolin-2(1H)-ylethanamine) and Gabapentin, capable of Aβ binding and inhibiting molecular aggregation, synaptotoxicity, intracellular calcium signaling, cellular toxicity and memory losses induced by Aβ. Thus, we recommend these molecules as novel candidates to assist anti-AD drug discovery in the near future. This review discusses the most recent research investigations about the Aβ dynamics in water, close contact with cell membranes, and several therapeutic strategies to remove plaque formation.

## 1. Introduction

Approximately 50 million people are globally affected by Alzheimer’s Disease (AD) [1,2]. This number will increase to 150 million by 2050 unless new prevention treatments become available [1]. Amyloid plaques and neurofibrillary tangles in brain tissue are the main hallmarks of AD. Amyloid plaques are composed of amyloid β (Aβ) peptides. Neurofibrillary tangles are composed of hyperphosphorylated tau proteins. In 1992, Hardy and Higgins [2] developed the amyloid cascade hypothesis; Aβ aggregates are transformed into Aβ fibrils that accumulate in the brain and finally trigger neurodegeneration. 

Amyloid precursor protein (APP) gene generates three variant APP695, APP751 and APP770, which are produced in neurons, endothelial cells, and platelets, respectively. In the aggregation pathway, transmembrane APP695 is cleaved by β- and γ-secretase to generates Aβ_1-40_ and Aβ_1-42_ peptides. During the APP695 cleavage, two processes occur (Figure 1). (1) Aβ is released to the extracellular hydrophobic environment, where Aβ monomers assemble into dimers, trimers, tetramers, oligomers, and fibrils [3,4,5,6], Aβ fibrils constitute the amyloid plaques considered a major pathological hallmark of AD. (2) A small amount of Aβ remains above the cellular membrane and can form membrane-associated Aβ oligomers that disrupt the shape of the membrane [7]. The amyloid fibrils are generally insoluble and transform into plaques. The Aβ oligomers are soluble and mainly spread throughout AD affected brain. Soluble Aβ oligomers deposited approximately 3–4 kDa in the AD brain could affect the calcium ion channel activity in synapsis through disrupting nerve signal transmission and damage mitochondrial causes to increase free radial lead to cell death. Soluble oligomers reach 10–100 kDa, which is considered more cytotoxicity than amyloid fibril aggregation; thus, the soluble oligomers exposed higher toxicity when compared with an insoluble fibril structure [8]. The water-mediated attraction in Aβ peptides and high propensity favor the formation of insoluble amyloid fibrils. The oligomers and fibrils conformation are recognized as a generic antibody epitope [9]. Although the relationship between oligomers and fibrils is still under debate, soluble and insoluble Aβ structures have been targeted to develop a cure for AD.

Molecular insight into aggregation pathways from oligomers to fibril formation remains an open problem in amyloidogenesis. A recent study reported [10] that during the aggregation, a large majority of oligomer structures are unstable and dissociate into their monomers instead of forming a new fibril structure, while the minority of the oligomers only convert into the fibril structure. Several research studies and hundreds of clinical trials since the early identification of the AD in 1906 have not been sufficient to discover an effective drug to alleviate the course of the AD disease [2], primarily due to the disordered nature of Aβ proteins that remains challenging for therapeutics. 

On 7 June 2021, the U.S. Food and Drug Administration (FDA) approved the aducanumab drug for mild AD patient treatment, which showed the removal of rich amyloid plaques and minimize side effects [11]. However, new drugs are required to cure AD completely. Many clinical trials with monoclonal antibodies targeting Aβ peptides have given negative results such as failure to remove rich plaques and produced severe side effects. As a consequence of Aβ and tau proteins triggered to decline cognitions of AD patients [12,13]. Thus, targeting tau protein rather than Aβ could be a promising approach to design novel drugs against AD. In this brief review, we discuss the following topics regarding the computer simulation efforts devoted by researchers in recent years: (a) development of methods and force fields (FFs) for the study of intrinsically disordered proteins (IDPs), (b) role of metal ions in amyloid peptide aggregation, (c) perturbation of Aβ peptide in membranes integrity and (d) design of inhibitors against AD.

## 2. Why Do Molecular Dynamics Simulations Cannot Accurately Quantify the A*β* Structural Ensemble?

Experimental studies have been unable to determine the properties of Aβ peptide in solution due to the fast conformational changes and enhanced aggregation tendency. These studies have produced time- and space-average results that are difficult to map into a conformational state of folded and unfolded proteins. Computational simulations can make a time series at the atomic level that could help us explore the protein structure, dynamics, misfolding and aggregation mechanism, becoming a particularly suitable complement to experimental studies of conformational changes of Aβ. Several force fields (FFs) to study biomolecules have been developed in the last decades, such as AMBER, GROMOS, OPLS families, namely AMBER94, AMBER96, AMBER99, AMBER99SBildn, AMBER03, AMBER12SB, AMBER14SB, CHARMM22*, CHARMM36, CHARMM36m, OPLS, GROMOS43a1, GROMOS43a2, GROMOS43a3, GROMOS53a5, GROMOS53a6 and GROMOS54a7. Most of the existing FFs describe phenomena associated with well-structured proteins. However, Saravanan et al. [14] concluded in a review study that the AMBER99SB-ILDN and CHARMM36m are highly optimized FFs and better choices for the characterization of IDPs such as Aβ peptide. This statement is supported because these FFs rendered the well agreement with experimental NMR chemical shift and β-sheet content, and the AMBER99SB-disp [15] force field is also worth considering for the same purpose.

Five recent FFs Amber ff14SB, Amber ff14SB_idps, Amber ff99SB, CHARMM36, CHARMM36m have been used by Pawel et al. [16] to explore the large conformational space of monomeric Aβ42 peptide during 10μs conventional molecular dynamics (MD) and 48 trajectories of replica exchange MD for 28.8μs. These FFs provided better results than their predecessor older versions. The potential energy can be described by *E*_total_
*= E*_bonded_
*+ E*_nonbonded_ where the bonded term (*E*_bonded_) consists of bond, angle, and dihedral-angle potentials, which explain the interactions of the atoms linked by covalent bonds, and the nonbonded term (*E*_nonbonded_) is constituted by van der Waals(vdW) and electrostatic interactions. The electrostatic and vdW components are the primary contribution to nonbonded energy for monomeric Aβ_1-42_. In the case of the CHARMM force field, the role of vdW interaction is reduced for Aβ_1-42_ peptide and enhanced for the Aβ_1-42_-water-ions interaction, whereas, in the case of Amber ff99SB, nonbonded potential energy slightly level up by the higher domination of electrostatic interaction, resulting in additional stabilization of the Aβ_1-42_ peptide related an over-structured β sheet. The interaction with water molecules contributes to the dynamics, misfolded and self-assembly of the Aβ peptide. The stronger solute-solvent interaction leads Aβ_1-42_ to be less stable and more hydrophilic. In addition, MD simulation studies with CHARMM36m and FF14SB_IDPs show antiparallel β-sheets between residues 16–21 and 29–36 of monomeric Aβ_1-42_, and short a β-strand in the C-terminal of the same monomer, which is in excellent agreement with NMR studies [17]. AMBER_ff14SB and AMBER_ff99SB overestimated α-helical and β-contents, respectively. Pawel et al. [16] strongly recommended using CHARMM36m force field for the study of the Aβ42-water-ion complex system over the AMBER FFs.

It is a big challenge to determine an accurate description of the structure of IDPs through MD simulations based only on FFs. In this perspective, Chong et al. [18] reviewed advanced computational methods that employ protein configuration entropy and render a thermodynamic connection between structural disorder and protein properties. For example, the CHARMM and OPLS FFs exhibit lower average β-sheet content in dimers of Aβ42 than that obtained with GROMOS 53a6 force field [18]. Subsequently, the average β-sheet content of the Aβ42 dimers was found to be greater in OPLSAA [19] than in AMBERFF99SB [20]. Interestingly, both AMBER99SB-ILDN and OPLS/L FFs have produced results of the average secondary structure of Aβ42 tetramer similar to each other [21]. 

The structural and thermodynamics properties of IDPs are susceptible to solute-solvent interaction compared to the folded protein. The choice of a reliable water model is necessary to characterize the Aβ42 peptide. Chong et al. [22] performed an MD simulation to investigate the structural properties of Aβ_1-42_ peptide by employing AMBER ff99SB force field with different solvent water models. They demonstrated that TIP4P-Ew exposed more Aβ_1-42_-water interaction than conventional TIP3P water model [18]. They strongly encouraged using the TIP4P-Ew water model to investigate the Aβ_1-42_ peptide structural properties. In a review, Chong et al. [18] reported that existing FFs are insufficient to expose Aβ protein to water. Recently, the same problem has been addressed by a couple of groups. The first group [23] opted to scale the Lennard-Jones potential between atoms in proteins and oxygen atoms in water by factor 1.1 without disturbing water-water and water-protein interaction. The second group [24] introduced a new water model, TIP4P-D, which included an additional parameter in the TIP4P water model to overcome the deficiencies in water dispersion interaction.

Recently developed FFs and their default water model are tabulated in Table 1. Rahman et al. [25] have evaluated the accuracy of recent developed FFs ff99IDPs, ff14IDPs, ff14IDPSFF, ff03w, CHARMM36m, and CHARMM22* by performing MD simulations for two short peptides (HEWL19 and RS), five IDPs (HIV-rev, Aβ_40_, Aβ_42_^1Z0Q^, Aβ_42_^model^, and pdE-γ) and two folded proteins (CspTm and ubiquitin) using trajectories of 1, 1.5, 5 or 10 μs for each system. They have compared J-coupling between MD simulation and NMR experiment for folded and disordered protein using different FFs. The J-coupling (J3-HNH_2_ and J3-HαC) parameter measures the secondary structure distribution based on φ backbone dihedral angle. Three IDPs FFs, ff99IDPs, ff14IDPSFF, ff14IDPs were in good agreement with the experimental J-coupling constant compared with tested FFs ff03w, CHARMM36m, and CHARMM22*. The balance between the local structural property (NMR chemical shift) and global structural property (Rg) is still a challenging issue for molecular simulations of IDPs. Rahman et al., [25] noted two observations: (1) average Rg for Aβ42^1Z0Q^ is 12.1 Å which is in close agreement with experimental Rg 12.4 Å, while ff03w showed Rg equal to 10.53 Å and CHARMM36m displayed Rg about 13 Å, which suggests highly divergence among those FFs; (2) the three IDPs FFs render a good balance between secondary structures contents for both Aβ_40_ and Aβ_42_^model^. While Amberff03w, CHARMM36m and CHARMM22* overestimated the α-helical structure for IDPs, thus favouring folded protein structures. Therefore, the three specific IDPs FFs were developed by incorporating the changes made in the pre-existing FFs (Table 1) to enable an accurate description of the folded and misfolded proteins [15,26,27,28,29,30,31,32,33,34,35].

The inconsistency of empirical physical models used in MD techniques can impact FFs and water models that affect the simulation result’s accuracy. Researchers have strived hard to develop perfect FFs to improve IDPs description; they aim to describe the high flexibility of these proteins, thus enlarging the conformational ensemble and increase the possibility of locating them in different local minima. Mu et al. [36] reported a couple of ideas to improve the accuracy of FFs for IDPs structural characterization, (1) Modification of force field parameters aided by global optimization, and (2) Maintaining a good balance between secondary structure via reparameterization (backbone dihedral parameters and vdW interaction between water-protein interaction) of existing FFs. One of the most common problems among the IDPs force field is over-stabilizing protein-protein interaction that impacts the aggregation mechanism of IDPs. Due to the IDPs force field’s inaccuracy, Mu et al. [36] encouraged improving backbone dihedral parameters and Lennard-Jones potential parameter (protein-water interaction) in the existing IDPs FFs and obtained training data from experimental observation and quantum chemical calculation. Undoubtedly, both reparameterization and training strategies may assist in new FFs development.

Another option to the atomistic FFs is to employ state-of-the-art coarse-grained (CG) models that not only sample more efficiently the entire space of protein conformations for large systems but also allow simulations for longer time scales of hundreds of µs [37], generally forbidden by brute force all-atom MD simulation and very relevant for biological processes [38] (e.g., folding, allosteric communications, conformational changes under mutations, self-assembly process, etc). Some popular CG FFs such as UNRES has been employed to study the fibril formations initiated by templates of Aβ_40_ fragments [39] capturing the dock-lock mechanism and similar the crowding effect of fragments was studied by PRIMO CG FF [40]. MARTINI 2 was employed to unveil the aggregation and organization of short Aβ16-22 peptides in lipid membranes [41]. The abovementioned CG FFs capture processes in time scales inaccessible atomic FFs, but yet they were restricted by system size or a lack of flexibility in secondary structure transitions. In this regard, the new release of MARTINI 3 force field aided by Gō-Like model [42,43,44,45] can become a new tool for realistic exploration of full-length Aβ peptide aggregation in contact with complex lipid-cholesterol membranes. Marrink group has made large efforts to develop a library of different lipid species (i.e., about 63 types) consistent with human plasma cells [46]. Following this idea, Poma et al. [47] have employed a very simplified CG model [48] to unveil the mechanical properties of Aβ40 and Aβ42 fibrils under different mechanical deformation process (e.g., tensile, indentation and shearing stresses) [49] and more recently they investigated [50] the change in mechanical stability between the oligomer and matured fibrils. The soluble oligomers are characterized by a length size about 3 to 5 nm, whereas Aβ fibrils typically reach hundreds of nm (see Figure 2). It is well-known the inverse relationship between toxicity and the length of the Aβ assembly [51]. Oligomers are considered more toxic than fibrils because of their high degree of flexibility toward low molecular weights, and the possibility of forming hydrophobic structures that may impair cell functions. Instead, fibrils are more thermodynamic stable and stiffer and less capable to undergo transitions to smaller and more toxic assemblies. Over the years, still some questions still remain open in terms of the mechanical characterization in amyloidogenesis and Aβ aggregate maturation: (a) what is the major role of the mechanical stability during Aβ oligomerization (e.g., tetramer, hexamer, etc)? (b) is the toxicity of oligomers which are closer to the membrane correlated by a minimum of mechanostability of Aβ peptide complexes, (c) how is the gain in mechanical stability from oligomers to fibrils involved in disease progression? and (d) Can we devise new strategies to reduce the mechanical stability of oligomer-to-fibril step through small molecular breakers (i.e., drugs recognition process)? To provide new answers to those open problems, new research combining versatile CG FFs and single molecule force spectroscopy is highly advised. For instance, in MD simulations, one can trace hydrophobic native interactions in equilibrium and simultaneously during a deformation process. Hence, the idea of hydrophobic structures can find support in molecular simulations as the main driving force for cell damage. Furthermore, molecular pathways could elucidate the critical conformation that maintains the mechanical stability of the Aβ assembly in the single-molecule force spectroscopy experiment. Figure 2 depicts the most relevant structure during aggregation and current methodologies used for its computational characterization [50,52].

Samantray et al. [35] have examined recently developed IDPs FFs, namely AMBER99SB-disp, CHARMM36m, and CHARMM36 with enhanced protein-water interactions (CHARMM36mW) for the study of Aβ_16-22_ (wildtype) aggregation and its mutation F19L Aβ_16−22_ (mutation 1) and F19 V/F20 V Aβ_16−22_ (mutation2), as model systems for testing purpose. In AMBER99-disp, the peptide-water interactions are increased too much resulting in an inhibition of the Aβ16-22 aggregation. The same trend has been observed in the simulation with the AMBERFF03w force field [53]. The difference between CHARMM36m and CHARMM36mW is a reparameterization of the protein-water interaction. In contrast, an experimental study [54] reported the following aggregation order mutation1>wildtype>mutation2≈0. AMBER99-disp does not apply for Aβ aggregation process because the interactions between peptide and water are drastically increased, leading to inhibition of the aggregation pathway. The CHARMM36mW can provide aggregation rate in the order of wildtype>mutation1>mutation2. Thus, FFs cannot reproduce the aggregation of Aβ peptide observed in experiments, but they maintain a better balance between peptide-water and peptide-peptide interaction. These results imply that improving the force field significantly impacted the simulation aggregation pathway than modifying the protein sequence. Samantray et al. [35] strongly encourage the use of CHARMM36mW for studying a full-length Aβ, even though it is not a perfect force field, since it yielded a promising result for aggregation benchmark. Nevertheless, reparameterization of this specific force field is still required.

Lockhart et al. [55] performed REMD simulations to examine the impact of the three popular FFs, CHARMM22 (protein FF) with CHARMM36 (lipid FF), CHARMM36m (protein FF) with CHARMM36 (lipid FF), and Amber14SB (protein FF) with Lipid14 (lipid FF), for the binding mechanism between the Aβ_10-40_ and the Dimyristoylgylcerophosphocholine (DMPC) bilayers. These three FFs have shown similar results in subjects like (a) stable helix formed in C-terminal of the peptide, (b) C-terminal of the peptide inserted into the bilayer hydrophobic core, (c) the thickness of the bilayer induced by the peptide about 10 Å and d) the disordered effect induced by the peptide on the fatty acid tails in the DMPC lipids. Nevertheless, these three FFs yielded different conformation ensembles of the peptide and bilayer that do not disturb the binding of the peptide with the bilayers.

Coskuner et al. [56] have reviewed several studies extensively and suggested that widely used FFs CHARMM, AMBER, GROMOS and OPLS, cannot produce accurate results for disordered entities. Even more, all existing computational techniques were designed to describe phenomena in ordered protein systems rather than in disordered protein structures. Important to mention that, Density Functional Theory (DFT) suffered from a number of errors that originated in the approximation of exchange-correlation functionals. These errors have been identified as the underestimation of barriers in describing chemical reactions, the band gaps, charge transfer excitation energies, and binding energies of charge transfer species in a biomolecule. DFT is the basis for constructing FFs such as CHARMM, AMBER, GROMOS and OPLS for intrinsically disordered protein and their complexes with ligands. These FFs are associated with an approximate exchange-correlation function that may lead to mistakes during prediction of the structural properties of IDPs. Notably, overcoming deficiencies of the exchange-correlation functional in DFT will help to improve the accuracy of IDP FFs. 

## 3. Why Does Aβ Peptide Perturb Membrane Integrity?

Three models [13] have been proposed for small protein insertion in lipid bilayers, (a) membrane pore formation model: Aβ oligomers formed within the membrane facilitates well-defined pores that lead to an unbalanced flow of ions in and out of the membranes, (b) carpeting model: Aβ peptide contacts the membrane surface to produce asymmetric pressure between two lipid layers of the membrane, which causes small molecules leakage, (c) detergent-like effect model: the lipids leave from the cellular membranes interacts with the Aβ peptide in resulting from perturbing membrane integrity. In this context, these models perturb the membrane through the interaction of Aβ peptide that yields to cell toxicity. Computer simulations assist in examining these models to unveil the molecular mechanism of the Aβ-membranes interaction. In general, three fundamental questions have not been elucidated yet, (1.) How does Aβ oligomeric intermediate perturb membrane integrity? (2.) What are the specific residues involved in membranes-Aβ interaction? (3.) What is the impact of metal ions in the mechanism of interaction between Aβ and membranes?

### 3.1. Aβ Monomer-Membrane

Although the Aβ peptides monomers were dissolved in non-polar environment, water solution and micelles environment using NMR techniques, there are no experimental studies on the 3D structure of monomers within the membrane environment [57]. In this perspective, computer simulations have been employed to unveil the interaction mechanism between truncated or full-length monomers and various types of membranes. For instance, MD simulations [58] subjected to examine the α-helical conformation (pdb id: 1Z0Q) and β-sheet conformation (pdb id: 2BEQ) of the Aβ_1-42_ peptide behavior on zwitterionic dipalmitoylphosphatidylcholine (DPPC) and anionic dioleoyl phosphatidylserine (DOPS) membranes, demonstrated that both membrane surfaces attract Aβ_1-42_ peptide, the attraction promotes dual mechanism, (a) the peptide perturbs the membrane surface and b) then Aβ_1-42_ follows aggregation by peptide-peptide interactions. Subsequently, REMD simulations were recruited to explore the same peptide in the same membranes [59], showing that no β-hairpin was found in the peptide except unstable β-hairpin on the anionic DOPS membrane, and the salt bridge Asp23-Lys28 gives significant contribution in the β-hairpin conformation of Aβ_1-42_ fibril and it is not formed in the membrane surface due to the Lys28 made electrostatic interaction with the charged lipid of the membranes. 

Lockhart et al., [60] have employed REMD simulations to study the binding between Aβ_10-40_ monomer and the DMPC bilayer in the presence of calcium ions. Results suggested two observations (a) the Asp23-Lys28 salt bridge is destabilized by the calcium ions and it is compelled to Lys28 interacts with the bilayers and (b) calcium ions reinforce the interaction between the monomer and the membranes by robust electrostatic interaction of charged amino acids to lipid polar head groups. These driving forces seemed to assisted the monomer in penetrating the lipid membranes.

In vitro studies reported faster aggregation of Aβ2_5-35_ compared to the full-length of Aβ peptide [61,62]. Smith et al. [63] probed the binding character of the Aβ_25-35_ and Aβ_10-40_ with the zwitterionic DMPC bilayer using REMD simulation and discovered two states when Aβ_25-35_ binding to bilayers, stable state-bound: peptide binding to surface polar head groups of the membrane, and less stable state: peptide inserted in the bilayer hydrophobic core. Free energy calculations confirm the Aβ_25-35_ transition between surface-bound and insertion state. In the case of the inserted state, the Aβ_25-35_ induces minor depletion in the lipid structure. In contrast, the C-terminal of the Aβ_10-40_ penetrates the bilayer deeply, inducing severe damage to membrane integrity, and thus, it confirmed the binding mechanism of Aβ_25-35_ and Aβ_10-40_ as different from each other.

### 3.2. Aβ Dimer-Membrane

The dimer of Aβ_17-42_ peptide insertion-membrane-pathway was examined using explicit solvent molecular dynamic simulations [64], with the structure extracted from the NMR data of Aβ_1-42_ fibril (PDB code: 2BEG) and with a U-shaped conformation and β-strand-turn-β-strand motif. The five different dimer configurations were generated, constructed the dimer on and in the lipid bilayer namely dimer1, dimer 2, dimer3, dimer4 and dimer5. Dimer1 was placed at the bilayer surface in the upper bilayer leaflet and is barely interacting with the bilayer leaflets. The dimer2 to dimer4 are partially embedded in the bilayers, and dimer5, fully immersed into the bilayers. The five different initial conformations of the dimers show different behavior on the DOPC (Dioleoyl, 1,2-dioleoyl-sn-glycero-3-phosphocholine) bilayers during the simulation. Jang et al. [64] have found that dimer structures reached equilibration more quickly in the membrane-embedded state than when dimers interact with the surface of the bilayer. Dimer1 slightly diffused into the bilayers with decreased beta-sheet contents, Dimer2 got inserted into the bilayer with modified initial conformation, Dimer3 managed to get inserted in the bilayer with loss of beta-sheet, Dimer4 was partially inserted with U-shaped conformation deviated significantly from initial conformation and finally, Dimer5 was deeply inserted into the bilayer preserving the U-shaped. Dimer 2 and 5 have maintained the U-shaped. These results demonstrated that U-shaped dimer is a more stable conformation observed both in solution and in the membrane. These structures are capable of penetrating the membranes that lead to the proposed mechanism of membrane toxicity.

Davis et al. [65] performed MD simulations to examine the dimer formation of Aβ_1-42_ peptide on zwitterionic DPPC and anionic DOPS membrane surfaces and found that both membranes promoted Aβ_1-42_ dimer formation. The DOPS membrane promotes the strong peptide-peptide interaction within the dimer exposed to solvent for the peptide aggregation. An opposite trend is followed by DPPC membranes in which peptide-peptide interaction becomes weaker during dimerization due to the strong interaction between peptide and lipid. These observations supported that DOPS rather than DPPC manifested the dimer process rapidly for Aβ_1-42_ peptide aggregation and in addition, DOPS served as a catalyst in the peptide aggregation.

Researchers have thought that two proteins, tau and amyloid, cause the AD disease for long decades. In contrast, Snowden et al. [66] proposed the hypothesis that Omega-3 and Omega-6 fatty acids are responsible for protective and pathogenic effects in AD. Lu et al. [67] have performed MD simulations to explore the structural character of Aβ_29-42_ dimer within the Omega-3 (docosahexaenoic acid) and Omega-6 (arachidonic acid) fatty acids membranes. Their results are compared with the Aβ_29-42_ dimer inserted into 1-palmitoyl-2-oleoyl-sn-glycero-3-phosphocholine (POPC) membrane system. MD simulations show that omega-3 membranes induced a higher population of the dimer disordered than in omega-6 membrane, mainly due to the decrease of beta-sheet and increase of helical content. Both fatty acid membranes yielded new conformations and orientation of the dimer compared to the POPC membrane. 

### 3.3. Aβ Trimer/Tetramer-Membrane

Aβ toxicity is significantly linked to its binding to the cell membrane and causes neuronal cell death. Jana et al. [68] have used mouse cortical neuronal culture to test the correlation between Aβ oligomer (up to tetramer level) binding and cell viability. Mouse cortical was treated with 5µm concentration of Aβ peptide for 96 h, Aβ42 peptide binding to neurons is 7-fold to 10-fold higher than that of Aβ_1-40_, which binds neurons after 24 h. Surprisingly, tetramers and trimers are observed after 48 h with less than 5% of total bound Aβ, and the peptide has not become toxic up to 48 h with 590 femtograms Aβ/cell bound. The peptide concentration was increased up to 15 µm and significant changes were observed when Aβ42 binding rate reached 777-923 femtogram Aβ/cell at 72 and 96 h, being neurotoxic triggered cell death and allowing trimers and tetramers to bind gradually to neurons, increasing over time for Aβ_1-42_ and contributing 15–20% of the total bound. The cell-bound trimer and tetramer (not monomer and dimer) play a vital role in Aβ_1-42_ toxicity (Figure 3). Small efforts exist on this matter, and researchers are highly encouraged to explore the Aβ trimer and tetramer peptides on the surface of membranes and within membranes using multiscale MD simulation to unveil cell toxicity mechanisms. 

A2V, H6R, D7N, A21G, E22Q, E22G, D23N mutations in Aβ peptides play a vital role to developing the disease in patients. On the other hand, A2T mutations protect patients from the AD because this mutated APP prevents the β-secretase cleavage [69]. We comment on some computational efforts about this topic. To gain knowledge of Aβ trimer neurotoxicity, Ngo et al. [70] carried out REMD simulations of Aβ_11-40_ trimer within DPPC lipid bilayer. They found that van der Waals interactions are the principal driving force for the binding between the trimer and lipid membrane, resulting in the penetration of the trimer in the bilayers. On the other hand, A21G [71] and F19W [72] mutations in the trimer decrease the presence of ASP23-LYS28 salt bridge, which leads to a destabilization of the trimer within the DPPC membranes. These mutations of Aβ_11-40_ trimers implied a lower aggregation tendency and reduced stability within the membrane compared to wild-type truncated trimer. The opposite trend is observed in the D23N mutant of the trimer. The D23 is a crucial residue as it forms Asp23-Lys28 salt-bridge in the loop region. The D23G mutation can reduce the total charge of oligomers, which causes a decrease in electrostatic energy of contacts between D23G in mutated Aβ_11-40_ monomers and promotes faster self-assembly and aggregation.

Arctic mutation (E22G) in Aβ_1-42_ peptides has been found in the human brain, enhancing Aβ_1-42_ aggregation, toxicity and playing a significant role in early-onset AD development. Poojary et al. [73] have tested the effect of Aβ_1-42_ mutations [74,75] in monomeric and tetrameric forms within the POPC membranes by using 500ns molecular dynamic simulations of native and mutated (E22G, D23G, E22G/D23G, K16M/K28M, and K16M/E22G/D23G/K28M mutants) Aβ_1-42_ peptide. The simulation results demonstrated that the monomeric and tetrameric structure of E22G mutated Aβ_1-42_ possesses higher stability, D23G mutant is the most unstable among the other studied peptides. Compared to other studied peptides, the tetramer of the D23G mutant exhibits an increased ability to perturb the membrane, which causes water permeation into the membrane. The pieces of evidence show that the tetramer of the D23G mutant has the highest toxicity, but the native and mutant monomer species are not toxic.

An early experimental study [76] failed to observe water molecules inside the Aβ fibril, but recent solid-state NMR studies [77,78] confirmed water molecules deep buried in the fibril which have been corroborated by MD simulations [79]. The difference between water molecule distribution in oligomers and fibrils remains an open question that deserves further exploration as water leakage has a crucial role in neurotoxicity. In this perspective, Nguyen et al. [21] address why the oligomers are more toxic than fibrils. They have identified the conformational space of Aβ_1-42_ tetramer (smallest stable oligomers) using coarse-grained FFs. They employed MD simulations to examine Aβ structural stability and compared them to NMR fibril structures. Thus, they proposed that the water density inside the oligomer is greater than in fibril, causing it to enhance the toxicity of oligomers. The interaction mechanism between tetramer and membranes interaction has yet been elusive, the open challenge question to the researcher is to investigate the smallest oligomer (Aβ_1-42_ tetramer) dynamic behavior on or within membranes (water-lipid interaction) by using computer simulations. This important piece of information will assist in unveiling neurotoxicity more quantitatively.

### 3.4. Aβ Oligomer-Membrane

Despite many efforts made in the amyloid pore hypothesis, none of the studies provides the atomic structure of Aβ oligomers. Without this information, it is impossible to elucidate the mechanism of Aβ oligomers toxicity. In this regard, Serra-Batiste et al. [80] examined the Aβ behavior within the membrane to elucidate the neurotoxicity in AD and found that Aβ_1-40_ and Aβ_1-42_ exhibited different physical behavior, Aβ_1-40_ aggregated into amyloid fibril while Aβ_1-42_ assembled into oligomers. The Aβ_1-42_ oligomers adopted β-barrel arrangement and formed well-defined pore into a membrane that named by β-barrel-pore-forming Aβ_1-42_ oligomers (βPFOS_Aβ42_). In comparison with Aβ_1-40_, Aβ_1-42_ has a significant role in AD by the higher propensity of βPFOS_Aβ42_. 

Carulla’s group further investigated [81] the atomic structures of Aβ_1-42_ tetramers and octamers within the membrane using experimental and MD simulations studies. The Aβ_1-42_ tetramer consisted of the six-stranded β-sheet core, Aβ_1-42_ octamers constituted by two Aβ_1-42_ tetramers, and both tetramer and octamers surrounded by a membrane. In the simulation study, the hydrophilic residues placed in both tetramers and octamers edges are unfavourably exposed to the hydrophilic lipid tails of the membrane. The head group of the membrane is reoriented towards hydrophilic edges of tetramers and octamers leading to stabilizing of the lipid-protein complex and resulting in lipid-stabilized pores. A high degree of water permeation is observed in the membrane and higher solvent accessible surface area in the octamers than tetramers. The water and ion penetration in the membrane was identified experimentally. Simulation and experimental results together proposed that water and ion penetration occurred through lipid-stabilized pores, mediated by hydrophilic amino acids in the β-sheet edges of the oligomers that could be responsible for neurotoxicity in AD.

Smith et al. [82] used replica exchange MD simulation for studying the aggregation of the Aβ_25-35_ peptide within the DMPC bilayer using CHARMM22/CMAP correction and CHARMM36 FFs for peptide and lipid FFs, respectively. In general, the Aβ_25-35_ peptide is characterized by two regions N-terminal (residues 25–28) and C-terminal (residues 29–35). The hydrophobic and electrostatic interactions between the Aβ_25-35_ dimers drives to produced parallel out-of-registry aggregation, which manifested pore formation in the bilayer. In particular, the aggregation within the membrane is constituted by three concentric rings, two outer rings located in the upper and lower leaflets are in contact through hydrophobic C-terminal with the bilayer core and N-terminal pointing towards the lipid head groups. The third rings are closer towards the pore centre, N-terminal directed the pore center to create pore lining. The peptide aggregation increases the extent of bilayer thinning that is more than four-fold more significant than the monomeric species; mainly because the dimers reduce the thickness of the DMPC bilayer from 40 Å to 24 Å. Thus, extensive damage to the membrane bilayers was mediated by uncontrollable Ca^2+^ permeation.

Conventional and steered MD simulations [83] have been employed to explore the zwitterionic POPC and palmytoil-oleoyl-phosphatidylethanolamine (POPE) head groups that influence the interaction between Aβ_9-40_ octamers and lipids. The results demonstrated that the POPC head groups form weaker electrostatic interaction with the Aβ octamer which have shorter-lived contact with the bilayers. Immediately, the head groups reorganized themselves, resulting in the lipid tail moving upwards to enhance electrostatic contact with the C-terminal of the octamer, which led to Aβ insertion into the membrane. This process is called a detergent-like effect on membranes for amyloid peptide formation. Whereas in the case of POPE-Aβ_9-40_ octamers, the head group repels the peptide insertion into membrane, this barrier is overcome by the stronger electrostatic interactions that persisted between charged residues in the Aβ_9-40_ and lipids bilayer and causing the C-terminal of the peptide to be inserted in the bilayer.

Qiang et al. [84] have used solid-state NMR spectroscopy to probed the phospholipids dynamics and interactions between lipid and peptide in a POPC bilayer fused with Aβ_1-40_ oligomers. At physiological conditions, lipids show changes in terms of lipid motion and reorganization. The stronger lipid-Aβ_1-40_ interactions restrict the lipid motion and inter-strand interaction between the loop and C-terminal of the oligomers. This effect is weakened by the lipid molecules inserted into oligomers that form rapid aggregates along with membranes intercalated by a hydrogen bonding network. The loop region of the oligomer could interact with the lipid head group to severely disturb membrane integrity by weakened interactions between lipid bilayers.

Fernández-Pérez experimental study [85] examines the relationship between cholesterol and Aβ oligomers behavior in a membrane. The membrane perforation induced by Aβ is much faster when a low concentration of cholesterol is present in the membrane. In contrast, a high concentration of cholesterol blocks the perforation effect of Aβ. Interestingly, neurons treated with cholesterol significantly increased Aβ association in the membranes compared with the free cholesterol. This evidence supports that the perforation effect of Aβ depends on the amount of cholesterol in the membrane and that cholesterol has a protective effect for Aβ toxicity. A large number of studies [13] provided evidence that cholesterol promotes Aβ aggregation and neurotoxicity.

Neurotoxicity of the Aβ has been elucidated by studies of the different forms of the Aβ peptide that interact with phosphatidylcholine (PC) and cholesterol. A dual mechanism has been observed [86], (a) The low concentration of cholesterol in the membrane interacts with Aβ peptide to promote the peptide insertion into the membrane and (b) opposing the peptide permeation at elevated cholesterol concentration through membrane stiffness effect. It gives evidence that Cholesterols manifested Aβ neurotoxicity [87,88,89,90,91]. In contrast, PC inhibits neurotoxicity by blocking aggregation and β-sheet formation [92,93].

The impact of cholesterol on the binding of Aβ (Aβ_17-42_ and Aβ_11-42_) fibrils with DPPC bilayers was investigated through MD simulations using Martini 2.0 force field [94]. The MD results supported the electrostatic interaction as the major driving force for the fibril binding to the membranes, along with the elevated level of cholesterol present in the membrane, which modulates this interaction by a dual mechanism, i.e., to the increased binding of positive residues with the lipid head groups and enhanced Ca^2+^ binding with the bilayers. The high concentration of cholesterol promotes Aβ_1-42_ peptide binding to the bilayer, in contrast, the cholesterol prevents the insertion of Aβ_10-40_ peptide into the bilayer. The relationship between Aβ peptide and membrane co-incubated with cholesterol is unclear, and computer simulations are still required to shed light onto this interaction.

## 4. How Do Metal Ions Govern Aβ Peptide Behavior?

The senile plaques are enriched by the presence of Zn^2+^, Cu^2+^, Fe^3+^ and Al^3+^ metal ions, it has been observed that in the case of a postmortem AD brain [95] their concentrations are 0.4 mM of Cu^2+^, 1 mM of Zn^2+^, and 1 mM of Fe^3+^; Al^3+^ has been also found in greater concentration in isolated core of senile plaques [96]. We will describe computational simulation efforts of metal interaction with Aβ peptides in the following sections.

### 4.1. Cu and Zn Ions Interactions with Aβ Peptides

In general, metal ion binding to peptide has been divided into three approaches, namely (a) bonded, (b) non-bonded, and (c) cationic dummy atomic models. Our previous research elucidated the dynamics of Aβ peptides upon metal interaction with bonded and non-bonded approaches [97,98]. For the non-bonded, we found Aβ_1-40_-Zn^2+^ and Aβ_1-42_-Zn^2+^ adopted a β-hairpin structure. For the bonded approach, the conformational space of Aβ_1-42_–Zn^2+^ is more heterogeneous compared to that of Aβ_1-42_–Cu^2+^ and Aβ_1-42_ because of the large number of basins present in the free energy surface (FES) of Aβ_1-42_–Zn^2+^ (see Figure 4). It confirmed that Aβ_1-42_–Zn^2+^ aggregates lead to a more amorphous state compared to other cases. In addition, we found that Zn^2+^ rather than Cu^2+^ binding promotes greater hydrophobicity in Aβ_1-42_. Our result agrees with Miller et al. [99] studies. They monitored the role of Zn2+ ions on Aβ oligomers by using MD and REMD simulations with the CHARMM27 force field. They observed that Zn^2+^ binding decreased the solvation energy (increase hydrophobicity) of Aβ oligomer, which enhanced the aggregation propensity, and that a higher concentration of Zn^2+^ could reduce aggregation kinetics. In contrast, the aggregation process became much faster than in metal-free solutions.

Lee et al. [100] performed MD simulations with the CHARMM27 force field and reported three significant insights: (a) Zn^2+^ ion mediated to increase the hydrogen bond network between Aβ_1-42_ layers of the oligomers, which caused to reinforce the stabilization of Aβ_1-42_ oligomers, in contrast, the same ion decreases the stabilization of Aβ_1-42_ fibrils by reducing the hydrogen bond network, (b) in comparison to Cu^2+^, Zn^2+^ destabilizes Aβ_1-42_ fibrils more effectively, Cu^2+^ does not reduce the hydrogen bond network in the fibril and (c) in comparison to Cu^2+^, Zn^2+^ binding did significant attenuate the energy of the salt bridge, which is an essential role in the formation of Aβ_1-42_ aggregation.

It is known that two residues are oxidized in Aβ, met35 is oxidized to a sulfoxide and Tyr10 is oxidized to form dityrosine covalent dimer; these oxidations impact the fibril formation. Met35 is oxidized by H_2_0_2_ alone without the presence of oxidation radicals. The oxidation of Tyr10 by radicals to form a Dityrosine covalent Aβ dimer has five- and eight-fold higher concentration in the hippocampus and neocortical region of the AD brain [101]. Cu^2+^ has a higher concentration in the AD patient brain, and H_2_0_2_ commonly exists at physiologically and mild acidic conditions. Miao et al. [102] have curiously monitored the relation between the concentration of Cu^2+^ + H_2_0_2_ and the amount of Dityrosine production by using experimental studies, noted 5 µm of Cu^2+^ concentration and a higher concentration of H_2_0_2_ from 0.4 mM to 1.6 mM that can enhance the amount of Dityrosine formation. By contrast, they found that the total dityrosine production is not affected by two-fold higher concentration Cu^2+^ ions (10 µm) with the same H_2_0_2_ concentration. These results supported H_2_0_2_ rather than Cu^2+^ as the main catalyzer in dityrosine productions. In addition, the Cu^2+^ +H_2_0_2_ redox cycling system oxidized most of the monomers within 100 h to produce a Dityrosine dimer. This effect accelerates the preformed Aβ_1-40_ fibril formation and reduces fibril length from 800 nm to 150 nm.

Santis et al. [103] utilized an X-ray Absorption spectroscopy to obtain the data for Aβ_1-42_ with Cu^2+^ and Zn^2+^ complex samples and reported that Zn^2+^ binding to Aβ_1-42_ could affect the Cu^2+^ coordination to the same peptide. If Zn^2+^ is added to the Aβ_1-42_ solution first, it prevents Cu^2+^ binding to the Aβ_1-42_ peptide. In contrast, if Cu^2+^ is added first does not inhibit Zn^2+^ binding to the Aβ_1-42_ peptide. 

In general, the Aβ peptide could not adopt a unique conformation because of their disordered nature. Its binding to metal ions is quickly is equilibrated in an aqueous environment compared with wild-type Aβ peptide. Srivastava et al. [104] answered the following fundamental question. Does metal ion affect the formation of aggregates and fibril formation? Of course, a high concentration of metal ions triggered fibril formation. For instance, Zn^2+^ and Cu^2+^ inhibit Aβ_1-42_ fibrillization and promote non-fibrillar aggregates. Later studies by Electron microscopy have shown that amorphous (non-fibrillar) aggregates convert into fibrils by binding metal ions. This result implies that amorphous aggregates are not the final point in the aggregation process. In other words, there are two cases, fibrils and small aggregation, for instance, metal ions are present in “amorphous” and “mature fibril” aggregates in which His6, His13, and His14 are coordinated with the metal ions. In particular, metal-bound His13 and His14 (beta-sheet conformation) side chains are found in opposite sides of the peptide. Later, the opposite trend followed by the small aggregation in which monomeric Aβ binds to metal ions through H6, H13, H14 and amino acid terminus, but H13 and H14 not bound with metal ions on opposite sides of the peptide chains resulting in around H13 and H14 regions could not convert β-sheet conformation. In some cases, His6 also does not turn into a β-sheet conformation in most of the fibrils since metal ions mediate a lesser binding effect. We commented that high concentrations of metal ions could mediate the amorphous Aβ aggregation, while low concentrations have triggered the Aβ fibril structure. It is noteworthy that the positive-charged metal ions can reduce the net-negative charge of Aβ, resulting in an enhancement of the aggregation rate of the peptide.

What metal ions induce Aβ cytotoxicity?

Aβ peptide can reduce Cu^2+^ to Cu^+^, and Fe^3+^ to Fe^2+^, facilitating the generation of reactive oxygen species H_2_0_2_ and OH^•^ radical. Tyr10 residue of the Aβ peptide loses one electron and produces reactive tyrosine radicals, which bind to the peptide to form a dimer, causing the killing of the brain cells [105]. On the other hand, (a) metal ions can induce conformational changes and are more aggregation-prone structures than the monomeric Aβ peptide and (b) they can allow via molecular bridging the formation of aggregates. 

To elucidate the role of Tyr10 residue on the Aβ self-assembly mechanism, Coskuner and Uversky [106] have utilized AMBER FF14SB and CHARMM22/CMAP FFs, employed REMD (7.2 μs) with thermodynamics calculation to study wild-type and Tyr10Ala mutation of Aβ_1-42_ monomer in the explicit aqueous environment. They have observed two issues a) Tyr10 residue promotes higher toxic β-sheet formation in the structural ensemble of Aβ_1-42_ monomer; however, b) an opposite trend followed by Tyr10Ala mutation decreases the β-sheet formation in the monomer. In general, the β-sheet formation in the peptide plays a vital role in self-assembly and fibril formation. This result revealed that Tyr10 regulates the toxic β sheet structure formation in the monomer causing faster self-assembly, but Tyr10Ala mutation impedes the self-assembly by decreasing the β sheet content. An open question is the impact of Tyr10 in Aβ in the presence of metal ions has not been elucidated yet.

Notably, the Cu^2+^ binding to Aβ generates less radial oxygen species in the case of murine than human Aβ, same in rats and mice that have not developed AD pathology because Aβ contained a lack of His13, a crucial residue in the binding of metal ions. The His13 residue mutation in human Alzheimer’s Aβ peptide is one of the ongoing research fields [107,108,109,110,111]. 

### 4.2. Fe Interaction with Aβ Peptide

So far, one study reported in the literature deals with Fe^2+^ interaction with Aβ peptide. We conducted 200 ns MD simulations with DFT study [112] to probe the structural dynamics of Aβ_1-42_-Zn^2+^, Aβ_1-42_-Cu^2+^ and Aβ_1-42_-Fe^2+^. Our results suggested that Fe^2+^ binding generates a U-shaped structure in Lys16-Met35 with the turn in the loop region and β-sheet extended over central hydrophobic and C-terminal regions, which is corroborated by the NMR study [76,113]. Overall, this result implies that Fe^2+^ binding enhances the fibril aggregation of the Aβ_1-42_.

### 4.3. Al Interaction with Aβ Peptide

Matthew Turner et al. [114] have employed five-microsecond MD simulation with AMBER ff14SB forcefield and study the process of how Al^3+^ governs the structural dynamic behaviour of Aβ_1-40_ and Aβ_1-42_ peptides. They observed Al^3+^ forming robust coordination with negatively charged E3, D7 and E11 residues, with coordination number of four. Such coordination persisted throughout the simulations despite metal ion bindings being modeled as a non-bonded model. This description has significantly impacted the structure and dynamics of both Aβ_1-40_ and Aβ_1-42_ peptides changes by promoting the helical structure formation by severely disrupting the salt-bridge network. Surprisingly, why is Al^3+^-Aβ binding coordination not disrupted throughout the simulation, even though a non-bonded metal binding model describes it? It is yet unclear. Is it an artifact of the chosen force field or simulation protocol that maintains the metal coordination? In this connection, Platts [115] utilized the semi-empirical tight-binding method GFN2-XTB for modelling the interaction of Al^3+^ with truncated Aβ_1-16_ peptide; the outcome of the optimized geometry is in agreement with DFT benchmark data. Metadynamics simulation has been used to explore exhaustively the coordination pattern of the same peptide [115]. The studies show that Al^3+^ binding to the Aβ_1-16_ is highly fluxional due to all acidic sidechains and several backbone oxygens involving coordination. The estimated coordination number on average is 5.2 atoms per ion. This evidence confirmed that the metadynamics approach does not match the microsecond MD simulation regarding the metal coordination number. The main reason is that in the case of non-bonded, atomic FFs could neglect charge transfer and polarization for peptide-ion interaction. In the bonded approach, atomic charges are obtained based on electrostatic potential (RESP) derived by B3LYP/6-31G (d) level of theory, amino acid backbone of oxygen to Al^3+^ is more electronegative −0.79 e as compared with MD simulations of non-bonded counterparts −0.40 to 0.58 e. These differences lead metadynamics to promote higher coordination number (5.2) than molecular dynamics coordination number (~4). MD simulation permits conformational changes of the Aβ without disrupting metal coordination, while metadynamics accounts for the different metal coordination mode.

Roldán-Martín et al. [116] have examined the structural features of Aβ_1-42_ peptide upon Cu^2+^ and Al^3+^ binding by using Gaussian accelerated MD simulation with AMBER FF14SB force field. They noted the following: (a) Cu^2+^ binding increased α-helical content in the Aβ_1-42_ peptide causes U-shaped conformations which is promoted by Glu22/Asp23-Ser26/Lys28 turn region that links two helices and arranges them in parallel through inter-helical hydrophobic contacts, (b) by contrast, Al^3+^ binding can convert the helical contents into the extended structure, which are more favourable for β-sheet formation, in other words, presence of Al^3+^ could have manifested aggregation-prone Aβ_1-42_ structures, and (c) the total charge of Aβ_1-42_ is neutralized by +3 charge of Al^3+^ metal, Aβ_1-42_-Al^3+^ complex becomes zero total charge, and the complex accelerates their aggregation propensity faster as compared to Aβ_1-42_-Cu^2+^ (charge, −1) and Aβ_1-42_ (charge, −3). 

### 4.4. Ag Interaction with Aβ Peptide

Wallin et al. [117] used Fluorescence spectroscopy, solid-state AFM imaging, circular dichroism spectroscopy, and NMR spectroscopy to probe the Aβ fibrillization pathways in the presence of silver (Ag^+^) metal ions. The Aβ peptide from the fibril incorporates other monomeric Aβ to elongate the fibril end. The Ag^+^ ion made weak binding to the N-terminal of the monomeric Aβ that lead to weakens the Aβ fibrillization kinetics by reducing fibril end elongation rate. Therefore, Ag^+^ bound peptide is insufficient of maintaining aggregation by decreasing the Aβ for fibril extension and the same observation is applied by the case Zn^2+^ binding with the Aβ peptide. These systems confirmed Ag^+^ crucial role in suffering the bulk aggregation by retardation of the fibril end elongation.

### 4.5. Pb Interaction with Aβ Peptide

The NMR studies confirmed the presence of Pb in the AD brain, this molecule induces an increase in the concentration of Aβ in rats [118] and enhances plaque formation in monkeys [119]. Wallin et al. [120] inspect the impact of the metal ions Cd^+^, Cr^2+^, Pb^2+^ and Pb^4+^ on Aβ aggregation by employed experimental studies. They found a couple of significant observations such as: (a) Pb^4+^ binds H6, Y10, E11, H13, and H14 residues of Aβ_1-40_ monomer and it can mediate faster aggregation rate, as compared with Cd^+^, Cr^2+^, and Pb^2+^ metal ions, which typically establishes electrostatic interaction with a monomer. The NMR and Fluorescence spectroscopy showed that Pb^4+^ binding to Aβ is slightly different from the Cu^2+^ and Zn^2+^ binding residues of D1, H6, H13, and H14, (b) Pb^4+^ binding induces M35 residue oxidation, facilitating the release of harmful reactive oxygen species (ROS) in the plaques that damage the neuron activities.

### 4.6. Hg Interaction with Aβ Peptide

Most humans are exposed to mercury by eating fish and shellfish contaminated with methylmercury. In general, Hg2+ ions cannot pass the blood-brain barrier (BBB), but metallic vapor mercury penetrated BBB to oxidized Hg^2+^ trapped inside the brain. Early studies [121] suggested the Hg^2+^ binds to thiol(-SH) and selenohydryl (-SeH) groups contained residue only. Thus, it does not bind the Aβ_1-42_ peptide because cysteine residues are absent. Nevertheless, recent experiments reported [122] Hg^2+^ directly interacting with Aβ peptide and strengthen the hypothesis of Hg^2+^ risk factor in AD. Subsequently, the molecular interaction of the Aβ_1-42_ with Hg^2+^ [123] has been explored using NMR, Fluorescence Spectroscopy and AFM, the outcome implies that Hg^2+^ binds to H6, H13, and H14 which can impede the Aβ_1-40_ and Aβ_1-42_ fibrillization and facilitate amorphous aggregation at a 1:1 Hg^2+^/Aβ ratio. 

### 4.7. Mn Interaction with Aβ Peptide

Several AD symptoms appear suddenly after severely exposed manganese [124], the relationship between Mn and AD is an ongoing debate. Wallin et al. [125] have used NMR, CD, AFM, Fluorescence spectroscopy and MD simulation to monitor the effect of Mn^2+^ ions on Aβ aggregation and fibrils. The NMR data [126] reported that Aβ_1-16_-Zn^2+^ complex with Zn^2+^ is coordinated by H6, H13, H14, and E11 residues. When Zn^2+^ manually was replaced by Mn^2+^ in the same complex and modelled Aβ_1-16_-Mn^2+^ complex with Mn2+ binds result in similar binding to the same residues: H6, H13, H14, and E11. This complex was used treated as an input structure for microsecond MD simulations. During MD simulation, the Mn^2+^ dissociated from initial binding residues, as it preferred to coordinate negatively charged residues of D1, E3, D7, and E11. These amino acids prevent the His residue from binding Mn^2+^ ion. 

Consequently, NMR data proved that D1, H13, D22, and E23 residues are more favourable for Mn^2+^ binding. Simulations and experiments predicted different binding sites for Mn^2+^, the exact details of Mn^2+^ coordination mode are still under investigation. Furthermore, CD spectroscopy and THT fluorescence data imply random coil (in the starting structure of Aβ peptide) transitions to β-sheets. In addition, the AFM image showed Aβ_40_ peptide aggregation mediates the formation of fibril structures, but Mn^2+^ does not influence at all on aggregation kinetics and fibril morphology of Aβ_40_ due to the weak interaction with the peptide. However, more simulations are required to model truncated Aβ_1-16_ or the full length of Aβ bound to Mn^2+^ ions.

Geng Lin et al. [127] have tested eight-month-old male mice divided into Mn-treated mice and controlled mice groups. Mn concentration was measured using mass spectroscopy at the end of the fifth-month treatment. The measurement ensured Mn level in similar concentration in blood and the brain. The qRT-PCR and western blot analysis confirmed in the case of Mn-treatment increased β- and γ-secretases cleavages at APP, an enormous amount of Aβ_1-42_ peptides production in the cerebral cortex and hippocampus of brains, while α-secretase cleavage activity was significantly reduced. These data suggested that Mn-treatment is a risk factor for the development of AD pathogenesis. However, no studies in the literature reported the effect of Mn on Aβ_1-42_ productions at atomic level details. The beforementioned studies supported the Aβ aggregation modified by Fe^2+^, Mn^2+^, Pb^4+^ and Zn^2+^, as well as the Hg^2+^ binding to specific residues of the Aβ peptide.

### 4.8. Li Interaction with Aβ Peptide

The AD symptoms are significantly reduced by the Li^+^ ions treatment [57,128,129], which showed a substantial improvement regarding Aβ clearance [130], enhance spatial memory [131], reduced oxidative stress level [132] and decreased the amount of Aβ plaques [133] in AD brain. Berntsson et al. [134] used biophysical techniques, NMR, Fluorescence quenching, CD, and IR, to monitor the elusive in vitro interaction between Li^+^ and Aβ peptide. They found that Li^+^ displayed weak interaction with the Aβ, resulting in unaltered secondary structures of Aβ_1-40_ monomers and Aβ_1-42_ oligomers. At elevated Li^+^ concentration, Li^+^ can manifest minor perturbation-effect on the morphology of aggregated Aβ_1-40_ fibril, Aβ_1-40_ aggregation kinetics and Aβ_1-42_ oligomeric stability. Overall, the results suggested Li^+^ ions are not able to modulate the Aβ aggregation and toxicity. The Li^+^ treatment on AD progression is not caused by direct interaction of Li^+^ with Aβ.

## 5. What Is the Role of Small Molecules in Inhibition Mechanism of Aβ Aggregation?

Researchers have widely used four novel therapeutic approaches to reduce plaque formation, (a) secretase inhibitors and modulators, (b) immunotherapeutic strategies, (c) peptide-based inhibitors, and (d) small molecular inhibitors. 

### 5.1. Secretase Inhibitors and Modulators 

Preventing the Aβ production by inhibiting the activity of β- and γ-secretase at APP has been a frontier biomedical research for AD treatment. In this account, Astra and Eli Lilly [135] have identified AZD3293 inhibitors that block the APP cleaved by β-secretase. Subsequently, several β-secretase inhibitors (OM99-2, KMI-429, GSK188909, 4-phenoxypyrrolidine, GRL-8234 and CTS-21166) were shown to attenuate plaque formation in the mice brain effectively [136]. The six γ-secretase inhibitors (DAPT, PF-3084014, LY-450139, BMS-708163, MPC-7869 and Begacestat) (Figure 5) have reached the clinical trial [136,137,138]. In 2001, an in-vivo study demonstrated for the first time that γ-secretase inhibitor DAPT (N-[N-(3,5-difluorophenacetyl)- l-alanyl]-S-phenylglycine t-butyl ester) blocked the Aβ production in the AD mice brain, but the lymphocyte development and the intestine symptoms were still observed [138]. During the prenatal period, Lymphocyte development occurs in humans. The new-born immune system contains functional T (thymus-derived) and B (born-marrow derived) lymphocytes. T and B lymphocytes are responsible for the function of antibody production and cell-mediated immune responses, respectively. Symptoms of intestine problems are stomach pain, vomiting, nausea, dehydration, a feeling of illness, and difficulty passing gas.

The PF-3084014 (Nirogacestat) has significantly reduced the Aβ level in the brain, plasma and cerebrospinal fluid in guinea pigs and Tg2576 transgenic mice. Since it was linked to a highly lipophilic nature, the clinical trial was discontinued due to an unfavorable pharmacokinetic and pharmacodynamic profile. The avagacestat (BMS-708163) significantly reduces Aβ_1-40_ levels in female rats, and significantly reduce Aβ_1-40_ in the brain and cerebrospinal fluid (CSF) of male beagle dogs [139]. The semagacestat (LY-450139) could reduce the plasma Aβ_40_ concentration by 58% and 65% for the 100-mg and 140-mg groups, respectively [140]. The avagacestat and semagacestat inhibitors blocked γ-secretase activity at APP in the cell membrane. However, these inhibitors failed in a phase 2 and phase3 of clinical trials because they generated a degeneration in cognitive ability in AD patients along with other severe side effects. 

During the semagacestat treatment, side effects were shown to increase the risk of skin cancer. Begacestat (GSI-953) decreased Aβ levels in plasma, brain, and cerebrospinal fluid in Tg2576 mice, but it failed to reduce Aβ concentration in the AD patient brain [141]. It was discontinued in the phase 1 trial in 2010. Tarenflurbil (MPC-7869) produced Aβ38 instead of Aβ42 by modifying the γ-secretase at APP. The Aβ38 manifested lower neurotoxicity and phase 2 trials showed encouraging results. The Phase 3 trial terminated in 2009 since the agent penetrate the blood-brain barrier insufficiently [137]. However, those inhibitors have decreased Aβ concentrations in the animal brain, but they showed severe side effects in AD patients.

### 5.2. Immunotherapeutic Strategies

In the immunotherapeutic approach, immunotherapy induces the host cell to recognize and fight Aβ in order to reduce Aβ aggregation in the brain. It is divided into two categories (a) active immunization and (b) passive immunization. The primary is to generate a vaccine for Aβ production to reduce plaques. The latter is to produce monoclonal antibodies and administration of immunoglobulins. The following five drug candidates, bapineuzumab, solanezumab, gantenerumab, aducanumab and Lecanemab, are humanized monoclonal antibodies except for solanezumab, which is another monoclonal antibody [142,143,144]. The first four drugs have significantly improved cognitive abilities and are undergoing clinical phase2 or 3 trials, while Lecanemab binds soluble Aβ and potential diseases attenuated by removing plaques.

The monomers, oligomers and fibrils of Aβ have exposed epitopes; typically, they are the same and antibodies can recognize and bind the epitopes of Aβ. Muller et al. [145] revealed that 20 amino acid SDPM1 proteins bind to the structure of the tetramers of Aβ40 and Aβ42 peptides, blocking Aβ association and removing plaques in AD brain. Solanezumab, Gantenerumab, and IVIG antibodies can improve cognitive skills by binding soluble Aβ peptides in the clinical trial [146,147]. The solanezumab and crenezumab have higher contact with the Aβ via KLVFF epitope [148]. The human anti-Aβ monoclonal antibody, gantenerumab, crossed the blood-brain barrier, binding aggregates of Aβ through the N-terminal and central regions of the peptide resulting in a removal of Aβ plaques, which was observed in phase 3 clinical study [143]. 

The drugs mainly target amyloid peptide, tau protein, mitochondrial dysfunction, oxidation stress and metal dysregulation to reverse the AD progression. Although a good amount (approximately 111 drugs) of drug candidates released between 2010 and 2020 [136], the majority of the candidates failed in phase 2 trials primarily due to the lack of the ability to reduce toxicity, cognitive loss and plaques formation, and more than 50 candidates reached phase 3 trial. 40% of the candidates fell into repurposed drugs. Notably, the four repurpose drugs, escitalopram, brexipiperazole, masitinib, and losartan, are undergoing phase 3 clinical trials. However, today, only one drug, aducanumab, successfully passed the phase 3 clinical trial and came to the market for mild AD patient’s treatment. 

The FDA approved two types of drugs (Figure 6a–d): (a) cholinesterase inhibitors and (b) memantine [1]. The primary can boost the amount of cholinesterase, which is responsible for memory functions. The latter inhibitor regulates the activity of glutamate, which is a messenger chemical involved in brain function, including learning and memory. There are three cholinesterase inhibitors: Donepezil (Aricept), Galantamine (Razadyne), and Rivastigmine (Exelon). Donepezil is used in all stages of AD patients, Galantamine serves to treat mild to moderate disease, and Rivastigmine in case of severe AD patients. These inhibitors reduce the symptoms of the disease with side effects including nausea, vomiting and diarrhea. Thus, these therapeutics can alleviate some of AD symptoms, but not the course of the disease.

In this perspective, Sevigny et al. [11] utilized positron emission tomography (PET) to monitor the efficacy of an aducanumab in mice and human brains. Aducanumab (Figure 6e) is an IgG1 monoclonal antibody and recombinant antibody derived from slow or absent cognitive decline patients. Three separated patient groups were subjected in this study, double-blind, randomized and placebo-controlled. The patients were clinically diagnosed with mild AD, and 1, 3, 6 and 10 mg kg^−1^ doses of aducanumab were given to 31, 32, 30 and 32 patients respectively for one year. In the end, the result was compared with the placebo-controlled group (40 patients). The PET measurement showed that the Aβ plaques formation has significantly reduced in subcortical white matter and cortical regions of the brain at 6 and 10 mg kg^−1^ within 54 weeks. In the placebo-controlled cases, no changes occurred at amyloid plaques. In addition, they gave 0.3, 1, 3, 10 and 30 mg kg^−1^ doses to mice for 9.5 to 15.5 months and discovered that 63% of Aβ plaques in the cortex and hippocampus of mice brains disappeared at 10 or 30 mg kg^−1^ doses. This evidence supports the strong effect of aducanumab entering the brain and reducing plaque formation by binding with the Aβ. All the data revealed aducanumab as a potential inhibitor for disease-modifying by decreasing soluble and insoluble Aβ [149,150]. Due to the removal of rich amyloid plaques and minimizing side effects, a milestone in AD therapy occurred on June 7th, 2021, the U.S. FDA approved aducanumab for the AD treatment [151].

Arndt et al. [152] deposited a crystal structure of AduFab (the fragment antigen-binding region of aducanumab)-Aβ1-11 peptide complex in the protein data bank (PDB id:6co3). In this structure, aducanumab binds to 3–7 residues of Aβ in an extended conformation. Furthermore, Frost and Zacharias [153] have performed MD simulation to explore the interaction between AduFab and Aβ1-40 peptide in the form of monomers, dimers, oligomers, and fibrils. The result suggested that AduFab binding affinity is increased in oligomers and fibrils compared to monomers.

### 5.3. Peptide-Based Inhibitors

Calcium ions flow in the membrane to increase the release of synaptic vesicles in hippocampal neurons that generate neurotransmission. In contrast, the Aβ plays a role in decreasing the number of synaptic vesicles that cause neurotransmission failure. Subsequently, the Aβ oligomer directly interacts with the membrane generating pore structure in which the flow of calcium is increased and leads to cell death. In this connection, Peter et al. [154] derived pentapeptide(G_33_LMVG_37_) from the glycine zipper region of C-terminal Aβ. Experiments characterized the Aβ activity upon the pentapeptide interaction, which confirmed G_33_LMVG_37_ involvement in three main activities: (a) blocking the increase of the calcium level in the neurons (b) inhibiting the perforation in the membrane (c) preventing the association of Aβ peptide. Finally, this pentapeptide supports neurotransmission by blocked Aβ induced membrane pore formation. In other words, the small size of the hydrophobic entity crosses the blood-brain barrier to reduce plaque formation. Other pentapeptides, G_25_SNKG_29_ and G_29_AIIG_33_, could not have any effect on Aβ peptide aggregation.

Zheng et al. [155] have employed multiple approaches such as ion mobility spectroscopy, mass spectroscopy, and MD simulation to characterize the interaction between the full length Aβ_1-42_ and two modified Aβ_39-42_ derivatives: VVIA-NH2 and Ac-VVIA. The mass spectroscopy revealed VVIA-NH2 binding to Aβ_1-42_ monomers, dimers, tetramers, hexamers, or oligomers, while Ac-VVIA only binds to monomers. Ion-mobility spectroscopy results showed that VVIA-NH2 prevents dodecamers formation and generates non-toxic oligomers, which form fibrils. On the other hand, Ac-VVIA mediated toxic oligomers eventually led to a fibril. MD simulation results suggested VVIA-NH2 has a weak binding affinity to the C-terminal region of monomers. In contrast, Ac-VVIA showed a strong binding effect to multiple regions of Aβ1-42. Overall, this data supported Ac-VVIA binding to oligomers as a crucial step for the inhibition of Aβ1-42 toxicity.

### 5.4. Small Molecular Inhibitors

Drug repositioning and repurposing are promising therapeutic strategies for drug discovery of anti-AD. Cramer et al. [156] have demonstrated that the anti-cancer drug, bexarotene, it can reduce 50% Aβ plaques in mice models of AD within 72 h. The following question is obscure: bexarotene destroys plaques by a direct interaction with Aβ. In this account, In this account, Pham et al. [157] employed in-silico and in-vitro experiments to inspect the role of bexarotene in the Aβ aggregation and reported that bexarotene showed week interaction affinity to Aβ fibril, which does not affect amyloid aggregation. The Aβ produced by the cleavage of APP by β- and γ-secretase, whose activity is tuned by the drug that is ongoing research to remove plaque formation. Subsequently, they also used in-silico and in-vitro experiments [158] to explore the possibility of preventing the Aβ association by bexarotene binding to β-secretase. The results imply that plaques could not be reduced due to the weak interaction between β-secretase and bexarotene. However, one more possible investigation remains obscure, examining the interaction mechanism between γ-secretase and bexarotene to understand the reduction of plaques. 

Mei et al. [159] have addressed the following question: How will the interaction between either negatively charged (ER) or neutral TS0/TS1 small molecules and Aβ42 oligomers affect the aggregation of oligomers? They have identified, in a comparative study with a small neutral molecule, ER (charged -2), as a suitable candidate to inhibit the neurotoxicity of Aβ_42_ aggregates through strong binding, decreasing aggregates hydrophobicity, degrading the β sheet contents and inter-and intra-molecular hydrogen bond on the main chain, and perturbation of K23-L28 salt bridges and the vdW interactions. Hence, they recommend developing negatively charged inhibitors for the target Aβ42 oligomers to prevent plaque formation.

As shown in Figure 7, the intrinsically disordered Aβ aggregation pathway has been characterized into four types [160]: (a) Primary nucleation, oligomers formed from monomeric species; (b) Elongation, oligomers and fibrils size increased by adding monomers; (c) Fragmentation, fibrils break into several oligomers and small fibrils that are capable of elongation; (d) Secondary nucleation, the surface of fibril aggregates promoting a new formation of oligomers species. Heller et al. [161] identified a small molecule, 10074-g5 (biphenyl-2-yl-(7-nitro-benzo [1,2,5] oxadiazol-4-yl)-amine), which interacts with the monomeric state of Aβ_1-42_ peptide using experiments, computational simulation and mathematical modelling. They observed (a) this molecule binds to monomeric Aβ_1-42_ peptide resulting in a significantly delayed the primary- and secondary-nucleation pathways in Aβ_1-42_ aggregation, (b) this molecule can decrease the hydrophobicity of the Aβ_1-42_ peptide and c) this molecule inhibits the formation of the Aβ_1-42_ association and toxicity by enhancing the conformation entropy.

Pagano et al. [162] have reviewed the role of the natural compound in the Aβ peptide aggregation pathways using in-silico and in-vitro studies. The thirteen compounds (Brazilin, Curcumin, Epigallocatechin gallate, Ginnalin A, wgx-50, Myricetin, Oleuropein, Oleuropein aglycone, Reserveratol, Rosmarinic acid, Sclerotiorin, Tanshinone and Uncarinic acid C) were extracted from plants and observed that they cross the blood-brain-barrier. Since the solvent exposure of the hydrophobic surface is regulated Aβ toxicity, inhibiting or modulating the toxicity of Aβ aggregation is one of the main approaches against AD. Considerable evidence coming from the experiments and computer simulations have confirmed that those thirteen compounds can directly interact with Aβ affecting amyloid aggregation pathway by (a) inhibiting formation of oligomers from monomers, (b) preventing the secondary nucleation of aggregation and (c) reducing the toxicity of aggregates.

Using REMD simulations, Mohamoudinobar et al. [163] have characterized the effect of NaCl, scylloinositol and 4-aminophenol in Aβ_1-42_ aggregation pathway in an aqueous environment. In the case of pure water, β-sheet is formed in the residue segments 16–22 (central hydrophobic region), 23–29 (central hydrophilic region) and the N-terminus, whereas in the case of NaCl contained water, the β-sheet is formed at N- and C-terminal, but the β-sheet formation in the central hydrophilic region is inhibited. Furthermore, in pure water, frequent residue contacts have been observed between the central hydrophobic and the C-terminus which were reduced by increasing the interaction between C-terminus and central hydrophilic region in the presence of inhibitors (scylloinositol and 4-aminophenol). Aβ_1-42_ peptide adopted a U-shaped conformation with strand-loop-strand structure in NaCl and pure water, but not in water-containing inhibitors. The U-shaped structure is well corroborated with the NMR fibril structure [164] and promotes Aβ_1-42_ aggregates. This result implies that the inhibitors prevent the Aβ_1-42_ aggregates, in contrast to the NaCl case.

In the past ten years of our research investigations, we have reached four main conclusions using in-silico, in-vitro and in-vivo experiments. (1) Nicotine molecule interaction causes the β-sheet of Aβ transformation into the alpha-helical structure that prohibits the aggregation of Aβ peptides [165]. (2) A venom of black mamba snake could inhibit the fibril formation of Aβ peptides by destroying the inter-peptide Asp23-Lys28 salt-bridge [166]. (3) U.S. FDA approved drug, gabapentin (GBP), binding to Aβ_1-42_ peptides and blocked multiple actions: aggregation, oligomers toxicity, membrane poration, intracellular calcium and synaptotoxicity induced by Aβ_1-42_ [167]. We recommend evaluating the GBP in a clinical trial for mild AD patients with a low dose of 400 mg/day to avoid risk associated with higher doses. (4) We identified a new low molecular weight compound M30 (2-Octahydroisoquinoline-2(1H)-ylethanamine) which generated a neuroprotective effect by removing the Aβ_1-42_ toxicity in hippocampal neurons and significantly improved spatial memory of AD affected mice [168].

## 6. Conclusion and Future Perspective

Why do researchers fail to identify drugs against AD? Although hundreds of molecules are present in the human brain, we believe Aβ and Tau aggregation are the main hallmarks of AD [169]. The hundreds of clinical trials targeting Aβ and Tau proteins have failed due to other molecules excluding Aβ and Tau proteins, which are invoking in initiating AD pathology. On the other hand, Aβ or Tau proteins interact with several biomolecules such as the membranes, transition metal ions, cholesterol, apoE3, apoE4, prion, Na+, K+, omega3, omega6, and other molecules. The experimental studies could not render those interactions. In addition, experiments have faced two main hurdles with predicting anti-AD drugs: unable to mimic the brain membrane exactly, prepared oligomers and fibrils structures are not resembling those presented in AD patients’ brains. They cannot provide the high-resolution structure of the Aβ oligomers. The atomic structural details of Aβ aggregates are in great need. Computer simulation studies are not only prepared based on the supplementary details of NMR experiments, but they also provide more insight on Aβ-Membrane and Aβ-drug interactions when experiments studies are challenging. In this account, it is highly encouraged to pursue new investigations using computer simulations.

Omega3 and Omega6 polyunsaturated fatty acids are buried in the neuronal membranes and are linked to a slowing down and increasing the risk of AD, respectively. Thus, it is essential to explore the dynamic mechanism of Aβ_1-40_/Aβ_1-42_ oligomers with and without interaction of metal ions in omega3 and omega6 phospholipids as these details are lacking in the AD research.Tyr10 residue in Aβ_1-42_ regulates the toxic β-sheet formation and leads to aggregation, but none of the studies in the literature regarding the role of Tyr10 in metal-bound Aβ_1-42_ peptide addressed this issue. In this perspective, investigating the interaction mechanism between metal-bound Aβ_1-42_ peptides and lipid membranes will be a crucial step to describe the toxicity of Aβ_1-42_ peptides.The inhibitory effect of the number of natural compounds in the aggregation pathways has been explained in the manuscript. However, their actions remain to be elucidated. The transition metal ions, Cu^2+^ and Zn^2+^, and their bindings induced a higher tendency of β-sheet formation in the Leu17-Met35 regions of Aβ_1-42_ peptide that can decrease solvent exposure (increase hydrophobicity) of the peptide which can lead to regulate toxicity [98]. To attenuate the hydrophobicity propensity of the peptide by the interaction of natural compounds, it is an essential investigation for anti-AD drug discovery.Large evidence [170,171] implicated the Aβ_1-42_ peptide aggregation controlled by lipid membranes, but in contrast, cholesterol in the lipid membrane significantly enhances the peptide aggregation by cholesterol, showing a high affinity for the peptide. However, the detailed interaction and its connection between the lipid bilayer and membrane remains elusive. The new idea is to study how free and metal ions bound to Aβ_1-42_ during aggregation changes in cholesterol-contained membranes.The Aβ_25-35_ cytotoxicity is well explained by the lipid bilayer mediated truncated Aβ_25-35_ peptide aggregation, which has made perforation in the membrane causing uncontrollable permeation of Ca^2+^ ions [82]. So far, exploring the full-length of Aβ_1-42_ peptide aggregation in the presence or absence of Cu^2+^, Zn^2+^ and Fe^2+^ metal ions in the membrane remains an open research problem. Here one can describe the cytotoxicity of the full-length Aβ_1-42_.Advanced MD simulation, aided by the improved version of coarse-grained force fields (e.g., MARTINI 3, UNRES, PRIMO, etc), can describe large-scale systems and overcome time-scale limitations, and improve our understanding of the connection between AD progression and initial stages of the disease by tracing the molecular pathways of full-length Aβ aggregation in complex with lipid bilayers. Similarly, it will help to unveil the fundamental role played by mechanical stability during amyloidogenesis and Aβ aggregate maturation.Our previous investigations have envisaged two molecules, M30 and Gabapentin, blocking multiple steps of the Aβ cascade: Aggregation, Synaptotoxicity, membrane pore formation, calcium dyshomeostasis and memory impairment. Thus, we strongly recommend using M30 and Gabapentin as part of the clinical trial as an alternative AD treatment.

## Figures and Tables

**Figure 1 ijms-22-10798-f001:**
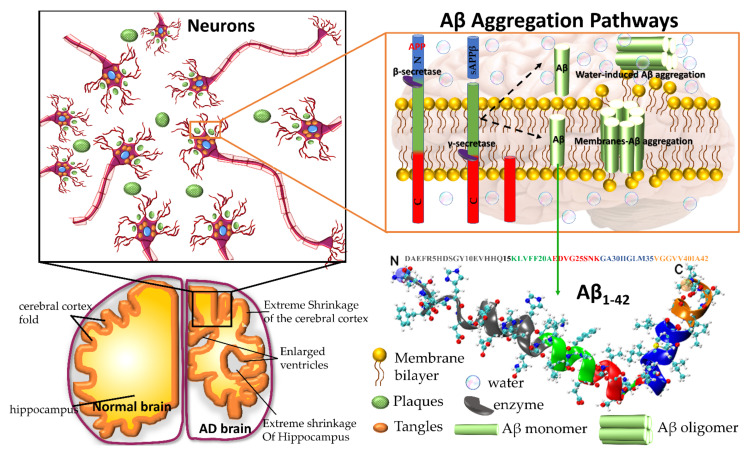
Plaques are around the neurons and tangles inside the neurons. Plaques and tangles are involved in killing the neurons resulting in drastic shrinking of the brain compared to normal brain. β-secretase cleaves APP (composed of 695 amino acids) into the membrane-tethered C-terminal fragments β (CTFβ or C99) and N-terminal sAPPβ. Aβ is obtained after the sequential cleavage of CTFβ by γ-secretases. Upon APP cleavage, two mechanism pathways have been proposed, in the first, Aβ is released to the extracellular environment, due to its hydrophobic nature, formed Aβ aggregates, in the second, a small amount of Aβ remains in the membrane evolving into membrane-associated Aβ aggregates. Aβ peptide has divided into five regions, N-terminus or hydrophilic region represent in grey, Central hydrophobic region in green, Loop region in red, second hydrophobic region in blue and C-terminal in orange. AD brain shrunken as compared with Healthy one.

**Figure 2 ijms-22-10798-f002:**
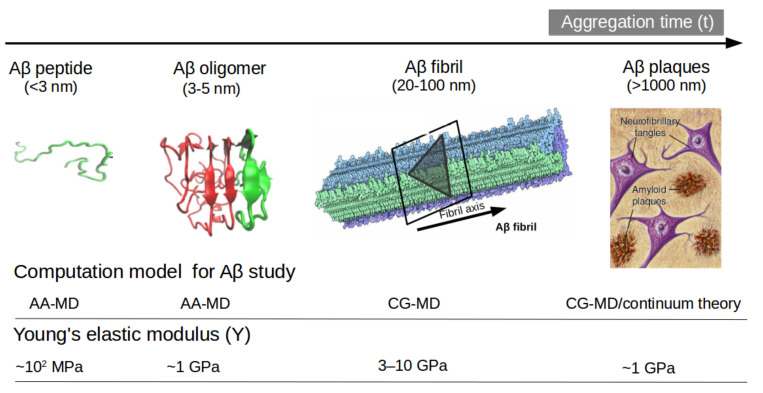
Representation of Aβ structures during aggregation: (a) free Aβ peptide, (b) oligomers, (c) fibrils and (d) plaques. Typical lengths are given for peptide (<3 nm), oligomers (3–5 nm) and fibrils (20–100 nm). All-atom MD (AA-MD), coarse-grained MD (CG-MD) and continuum models have been employed to unveil the mechanical stability for each structure. Young modulus (Y) defined by the ratio of applied tensile stress to a given strain provides an idea of the elastic regime. Y values for each structure is taken from ref (50 and 51). Some images are modified with permission of BrightFocus Foundation.

**Figure 3 ijms-22-10798-f003:**
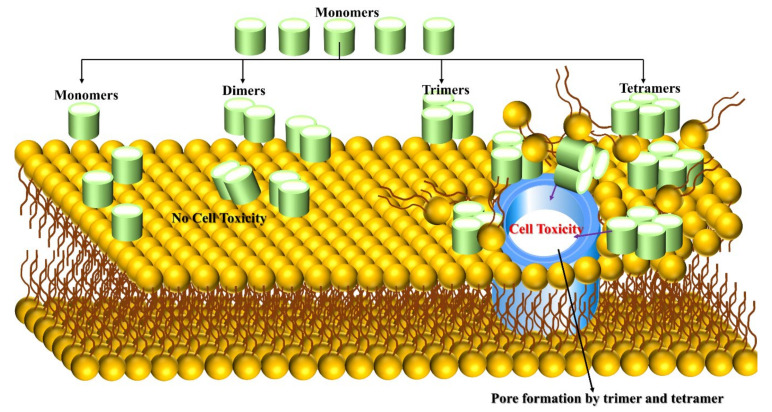
Aβ monomers and dimers interacting with membrane bilayers do not cause cell toxicity but tetramers followed by trimer binding causes the cell toxicity.

**Figure 4 ijms-22-10798-f004:**
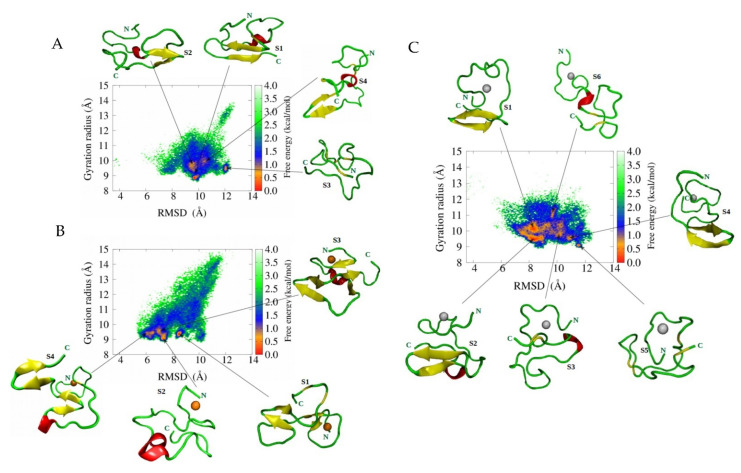
Free energy landscape for A. Aβ42, B. Aβ42-Cu^2+^, and C. Aβ42-Zn^2+^ peptides as a function of RMSD and the gyration of radius. Results are based on the whole ensemble of trajectories. Representative structures in the free energy minimum basins are displayed. Reproduced with permission from the work of Boopathi et al. [98]. Copyright 2020 John Wiley and Sons.

**Figure 5 ijms-22-10798-f005:**
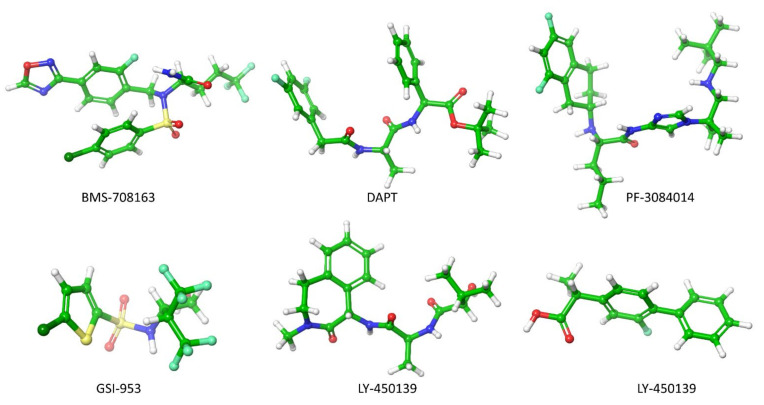
The γ-secretase inhibitors used in clinical trial for the treatment of AD and taken from PubChem.

**Figure 6 ijms-22-10798-f006:**
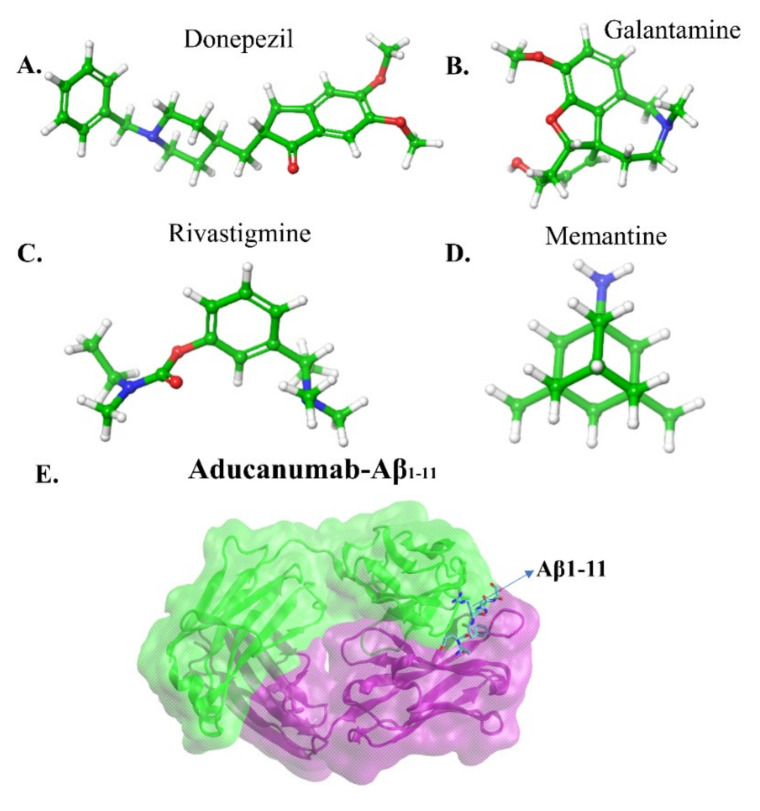
The U.S FDA approved drugs for AD (**A**) Donepezil, (**B**) Galantamine, (**C**) Rivastigmine and (**D**) Memantine. (**E**) fragment antigen-binding region of aducanumab (AduFab) -Aβ_1-11_ peptide. AduFab depicted by van-der-Waals surface. The chemical structures taken from PubChem and protein data bank.

**Figure 7 ijms-22-10798-f007:**
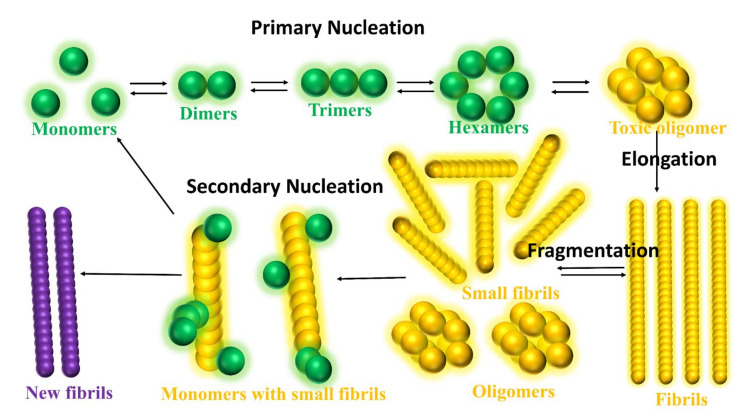
Schematic illustration of the aggregation pathways. There are characterized into four types: Primary nucleation, monomers formed dimers, trimers, hexamers and then oligomers. Elongation, the oligomers are joined together to generated the fibrils. Fragmentations, the fibrils break into the number of small fibrils and oligomers. Secondary nucleation, only minority of small fibrils convert into a new fibril by adding monomers, but due to the addition of monomers, the majority of the small fibrils turn into the monomers or oligomers.

**Table 1 ijms-22-10798-t001:** Latest developed force field for intrinsically disordered protein and water model.

Force Field	Parameter Set	Developments	Water Model	Reference
AMBER	ff99IDPs	Updated from ff99SBildn by adding a set of backbone torsion parameters of eight disordered promoting amino acids.	TIP3P	Wei Y et al. [26] 2015
	ff14IDPs	Updated from ff14SB by embedding a set of backbone torsion parameters of eight disordered promoting amino acids.	TIP3P	Song et al. [28] 2017
	ff14IDPSFF	Updated from ff14SB by introducing a set of backbone torsion parameters for 20 amino acids	TIP3P	Song et al. [29] 2017
	ff03CMAP	Updated from ff03 by introducing a correction maps (CMAP)-optimized force field	TIP4PD (Modified the dispersion interaction of the TIP4P)	Zhang et al. [30] 2019
	ff14SB	Updated from ff99SB by improving the Accuracy of Protein Side Chain and Backbone Parameters	TIP3P	Maier et al. [31] 2015
	ff03w	Updated from ff03 by adding slight backbone modification	TIP4P/2005	Best et al. [32] 2010
	A99SB_disp	Update from a99SB-ILDN by an introducing small change in the protein and water vdW interaction terms	TIP4P-D	Robustelli et al. [15] 2018
CHARMM	CHARMM36m	Updated from CHARMM36 by a refined backbone correction map potential	CHARMM-modified TIP3P	Huang et al. [33] 2017
	CHARMM36IDPSFF	Updated from CHARMM36m by CMAP corrections made for all 20 naturally occurring amino acids	CHARMM-modified TIP3P	Liu H et al. [34] 2019
	CHARMM22*	Updated from CHARMM by introducing modifications in backbone torsion potential	CHARMM-modified TIP3P	Stefano Piana et al. [35] 2011
	CHARMM36mW	Van der Waals interaction between protein and water are included in CHARMM36m	CHARMM-modified TIP3P	Samantray et al. [27] 2020

## Data Availability

Not applicable.

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
