# Peer review of "An Overview of Several Inhibitors for Alzheimer’s Disease: Characterization and Failure"

_ijms, 2021, doi:10.3390/ijms221910798_

Round 1

Reviewer 1 Report

A Computational Perspective: Why Do Researchers Fail to Identify Drugs Against Alzheimer's Disease?  By Subramanian Boopathi, Adolfo B. Poma, and Ramón Garduño-Juárez

This review manuscript by Boopathi et al. summarizes recent advances in our understanding of amyloid-beta aggregation and cell toxicity and degeneration. It includes several computational and experimental studies, with a comprehensive discussion of molecular and cellular details, and some mechanisms of drug action and development.

Although the reader is likely to obtain novel information in a unique fashion from this manuscript, several aspects can be improved as describe below:

  1. There is a figure next to the abstract that has no legend explaining what it is showing, and it is not mentioned in the text.
  2. For figure 1 the aggregation panel could use more space on the figure, so it appears less crowded. The legend should explain whether oligomer or fibrils are depicted and what the circles (probably water molecules) represent. It is unclear what the panel showing healthy vs. AD brain is showing. The panel with the neuron seems too large, is not included in the legend, and indicates that the A-beta plaques are only located at the dendrites. This should be clarified in the legend.
  3. The sentence on lines 45-46 is confusing, as it sounds like tangles are also composed of A-beta
  4. The paragraph in lines 50-62 is an interesting one, but it should clearly explain how fig. 1 shows each of the two aggregation pathways described in the text, and which one leads to fibril formation.
  5. The sentence in lines 81-82 is confusing, as it is unclear whether the “consequence” is the explanation of why the clinical trials showed negative results
  6. Several abbreviations require explanation the first time they are included in the text (for example IDP, MD, DPPC, wt, m1, m2, etc.)
  7. Some panels in the figures include figures that have been used in previous publications (for example A-beta plaques on fig. 2, or the entire figs. 4 & 6). The authors need to make sure that they have permission to use them here as well.
  8. The legend of fig. 3 should clearly state that it is only showing one of the three proposed models for small protein insertion in lipid bilayers, and why this is the supported model.
  9. The paragraph in lines 320-332 is an interesting one, but it should better explain how the five different conformations/dimers were established.
  10. In line 345, the authors mention how Snowden et al. used clinical trials to propose that tau and amyloid are less linked to memory problems, but Omega-3 and Omega-6 fatty acids would play a more relevant role. Considering the whole review focuses on A-beta peptides, the authors should explain why it is still relevant to study A-beta peptides, and if there are any limitations on studying Omega 3 or 6 (in other words, why is this review not on omega 3 & 6 if they are really more important molecular players?).
  11. The discussion about mutations in lines 376-401 is an interesting one. The reader would benefit from a better clarification of which mutations have been found in AD patients. Is the high toxicity of D23G also observed in human brains?
  12. In line 215 the authors asked the question “is the gain in mechanical properties from oligomers to fibrils responsible for disease progression?”, but then in line 407 it is said: “the oligomers are more toxic than fibrils”, which seems contradictory. A clear discussion of toxicity from oligomers vs. fibrils should be addressed earlier in the text (probably around the same time when figure 1 is introduced), so the reader gets a realistic image of how these structures compare to each other.
  13. The discussion about the role of cholesterol in lines 476-490 is an interesting one. However, it seems confusing for the reader and it seems not easy to conclude whether cholesterol has protective properties or if it promotes neurotoxicity.
  14. The sentence in line 513 seems incomplete
  15. In line 557 it is unclear what is being compared to the wild type A-beta peptide (unless what is meant with “the wild type A-b peptide” is the absence of metal ions ?)
  16. The question asked in line 559 is an interesting one, but I’m not sure the answer should be presented as a clear “of course”
  17. In line 739, it would be beneficial to further explain what the lymphocyte development and the intestine symptoms are
  18. Since the title of the manuscript says “Why Do Researchers Fail to Identify Drugs Against Alzheimer's Disease?” the reader would benefit from additional ideas to answer this question in the conclusion section. The only idea presented here (“have failed due to other molecules excluding Aβ and Tau proteins”) does not seem to provide enough depth to the reader.
  19. It would be beneficial to clarify how computer simulations would allow the examination of detailed molecular interactions, which, in turn, those predicted interactions would need to be confirmed by subsequent experimental studies.
  20. The way some of citations are included in the first pages of the text is not consistent and misleading (for example on line 44 “million by 20501 unless”).

Author Response

Dear Reviewers,

Our manuscript has greatly benefited from your comments, and now reinforced the quality of the manuscript. We would like to thank you for sparing precious time and effort for reading the manuscript and providing insightful comments. We have provided point-by-point responses to your comments. The changes made in the manuscript has been highlighted in yellow.

First Reviewer’s Comments:

This review manuscript by Boopathi et al. summarizes recent advances in our understanding of amyloid-β aggregation and cell toxicity and degeneration. It includes several computational and experimental studies, with a comprehensive discussion of molecular and cellular details, and some mechanisms of drug action and development. Although the reader is likely to obtain novel information in a unique fashion from this manuscript, several aspects can be improved as describe below:

Question 1: There is a figure next to the abstract that has no legend explaining what it is showing, and it is not mentioned in the text.

Response: We provided a legend explanation for this figure. We could not give this information into the text because it is an abstract figure.

Question 2: For figure 1 the aggregation panel could use more space on the figure, so it appears less crowded. The legend should explain whether oligomers or fibrils are depicted and what the circles (probably water molecules) represent. It is unclear what the panel showing healthy vs. AD brain is showing. The panel with the neuron seems too large, is not included in the legend, and indicates that the A-β plaques are only located at the dendrites. This should be clarified in the legend.

Response: Yes, according to your suggestion, we have modified the figures.

Question 3: The sentence on lines 45-46 is confusing, as it sounds like tangles are also composed of Aβ

Response: Amyloid plaques and neurofibrillary tangles in brain tissue are the main hallmarks of AD. Amyloid plaques are composed of amyloid β (Aβ) peptides. Neurofibrillary tangles are composed of hyperphosphorylated tau proteins.

Question 4: The paragraph in lines 50-62 is an interesting one, but it should clearly explain how fig. 1 shows each of the two aggregation pathways described in the text, and which one leads to fibril formation.

Response: Water-mediated attraction propensity dictates the soluble Aβ peptides form an insoluble amyloid fibril.

Question 5: The sentence in lines 81-82 is confusing, as it is unclear whether the “consequence” is the explanation of why the clinical trials showed negative results.

Response:  Many clinical trials with monoclonal antibodies targeting Aβ peptides have given negative results such as failure to remove rich plaques and produced severe side effects. As a consequence of Aβ and tau proteins triggered to decline cognitions of AD patients. Thus, targeting tau protein rather than Aβ could be a promising approach to design novel drugs against AD.

Question 6: Several abbreviations require explanation the first time they are included in the text (for example IDP, MD, DPPC, wt, m1, m2, etc.)

Response: Abbreviations have been provided the first time we mentioned them in the text.

Question 7: Some panels in the figures include figures that have been used in previous publications (for example fig. 2, or the entire figs. 4 & 6). The authors need to make sure that they have permission to use them here as well.

Response: Yes, we have received permission to use those figures in our manuscript.

Question 8: The legend of fig. 3 should clearly state that it is only showing one of the three proposed models for small protein insertion in lipid bilayers, and why this is the supported model.

Response: This figure did not describe the three models: carpeting model, membrane pore formation model and detergent-like effect model. However, it mainly elucidated the tetramers and trimers mediated the neuronal cell death by extensive damage to the cellular membranes via forming pore structures. Nevertheless, monomers and dimers structures were not involved in a significant disruption of the membrane integrity and cell toxicity.

Question 9: The paragraph in lines 320-332 is an interesting one, but it should better explain how the five different conformations/dimers were established.

Response:  The Aβ17-42 dimer structure was extracted from the solid-state NMR studies of Aβ17-42 fibrils (PDB id: 2BEG), five different dimer configurations were generated and constructed the dimer on and in the lipid bilayer.  The dimer1 was placed at the bilayer surface in the upper bilayer leaflet and is barely interacting with the bilayer leaflets. The dimer2 to dimer4, partially embedded in the bilayers, and dimer5, fully immersed into the bilayers. They found that dimer structures attained equilibration quickly in the membrane-embedded state of dimer than dimers interact with the surface of the bilayer.

Question 10: In line 345, the authors mention how Snowden et al. used clinical trials to propose that tau and amyloid are less linked to memory problems, but Omega-3 and Omega-6 fatty acids would play a more relevant role. Considering the whole review focuses on A-β peptides, the authors should explain why it is still relevant to study A-β peptides, and if there are any limitations on studying Omega 3 or 6 (in other words, why is this review not on omega 3 & 6 if they are really more important molecular players?).

Response: We have modified the text in the manuscript according to your suggestion.

Question 11: The discussion about mutations in lines 376-401 is an interesting one. The reader would benefit from a better clarification of which mutations have been found in AD patients. Is the high toxicity of D23G also observed in human brains?

Response: Arctic mutation (E22G) in Aβ42 peptides has been found in the human brain, enhancing Aβ42 aggregation and toxicity and playing a significant role in early-onset AD development. D23G mutation at tetramer structure can affect the polarization of the membranes leads to membrane structural integrity collapsed, resulting in formation of small molecules leakage. Thus, these mutations quite dangerous compared other mutation. Yes, it can induce higher toxicity, among other mutations.

Question 12: In line 215 the authors asked the question “is the gain in mechanical properties from oligomers to fibrils responsible for disease progression?”, but then in line 407 it is said: “the oligomers are more toxic than fibrils”, which seems contradictory. A clear discussion of toxicity from oligomers vs. fibrils should be addressed earlier in the text (probably around the same time when figure 1 is introduced), so the reader gets a realistic image of how these structures compare to each other.

Response: Thanks for the opportunity to clarify our text. We have added a discussion of the toxicity between oligomers and fibrils in the main text, and we reformulated the question to make clear the message.

We have addressed the following information in the introduction section. Soluble Aβ oligomers deposited approximately 3-4 kDa in the AD brain could affect the calcium ion channel activity in synapsis through disrupting nerve signal transmission and damage mitochondrial causes to increase free radial lead to cell death. Soluble oligomers reached 10-100 kDa is considered more cytotoxicity than amyloid fibril aggregation.

Question13: The discussion about the role of cholesterol in lines 476-490 is an interesting one. However, it seems confusing for the reader and it seems not easy to conclude whether cholesterol has protective properties or if it promotes neurotoxicity.

Response:  A large number of studies provided evidence that cholesterol promotes Aβ aggregation and neurotoxicity. It was added in the manuscript.

Question14: The sentence in line 513 seems incomplete

Response: We have improved the sentence. For the bonded approach, the conformational space of Aβ1-42–Zn2+ rather than Aβ1-42–Cu2+ and Aβ1-42 is more heterogeneous because of the large number of basins present in the free energy surface (FES) of Aβ1-42–Zn2+ as shown in Figure 4. It confirmed that Aβ1-42–Zn2+ aggregates lead to more amorphous compared to other cases.

Question15: In line 557 it is unclear what is being compared to the wild type A-β peptide (unless what is meant with “the wild type A-b peptide” is the absence of metal ions?)

Response: As the sentence is not clear, we removed it.

Question 16: The question asked in line 559 is an interesting one, but I’m not sure the answer should be presented as a clear “of course”

Response: High concentrations of metal ions can mediate the amorphous  aggregation, while low concentrations have triggered the Aβ fibril structure. It is noteworthy that the positive-charged metal ions can reduce the net-negative charge of Aβ, resulting in an enhance the aggregation rate of the peptide.

Question 17: In line 739, it would be beneficial to further explain what the lymphocyte development and the intestine symptoms are

Response: During the prenatal period, Lymphocyte development is occurred in humans. The newborn immune system contains functional T (thymus-derived) and B (born-marrow derived) lymphocytes. T and B lymphocytes are responsible for the function of antibody production and cell-mediated immune responses, respectively.

Symptoms of intestine problems are stomach pain, vomiting, nausea, dehydration, a feeling of illness, and difficulty passing gas.

Question 18: Since the title of the manuscript says “Why Do Researchers Fail to Identify Drugs Against Alzheimer's Disease?” the reader would benefit from additional ideas to answer this question in the conclusion section. The only idea presented here (“have failed due to other molecules excluding Aβ and Tau proteins”) does not seem to provide enough depth to the reader.

Response: Experiments have faced two main hurdles with predicting anti-AD drugs: unable to mimic the brain membrane exactly, and prepared oligomers and fibrils structures are not resembling those presented in AD patients’ brains. They cannot provide the high-resolution structure of the oligomers of Aβ peptides, and the atomic structural details of Aβ aggregates are in great need.

Question 19: It would be beneficial to clarify how computer simulations would allow the examination of detailed molecular interactions, which, in turn, those predicted interactions would need to be confirmed by subsequent experimental studies.

Response: Additional information about the role of MD simulation for the characterization of specific interactions is given in the context of the mechanical stability and Single Molecular Force Spectroscopy experiment.

Question 20: The citations included in the first pages of the text is not consistent and misleading (for example on line 44 “million by 20501 unless”).

Response: The citation has been inserted in the text.

Reviewer 2 Report

This manuscript includes an imposing review of the studies on the Alzheimer disease (AD) performed so far especially focusing on the computational works. The authors give some future directions to solve the various problems to tackle and develop drugs against the Alzheimer disease in the end of this review. This review will be very useful for the many researchers of AD. Thus, I feel that this review is worth publishing but also the following points should be reconsidered before publication.

I think that the title of this review is slightly unsuitable because this review includes various aspects of the AD besides computational works. The title should be changed to emphasize that this review treats various aspects of AD. For example, in the conclusion sections, the explicit point related to the computation is just 6.

The minor points;

  1. In page 1, line 43, should “20501” be “2050”?
  2. The citation of a literature should be indicated by a superscript number.

For example, in page 2, line 71, “reported10”.

Author Response

Second reviewer comments

This manuscript includes an imposing review of the studies on Alzheimer disease (AD) performed so far especially focusing on the computational works. The authors give some future directions to solve the various problems to tackle and develop drugs against Alzheimer disease at the end of this review. This review will be very useful for the many researchers of AD. Thus, I feel that this review is worth publishing, but also the following points should be reconsidered before publication.

Question 1: I think that the title of this review is slightly unsuitable because this review includes various aspects of the AD besides computational works. The title should be changed to emphasize that this review treats various aspects of AD. For example, in the conclusion section, the explicit point related to the computation is just 6.

Response: We appreciate the concern of the reviewer. However, we consider the title given in the review article in agreement with the scope of the study. We tried to use the available experimental information to path the way for new simulations that may lead to new discoveries.

 Question 2: The minor points; On page 1, line 43, should “20501” be “2050”? The citation of literature should be indicated by a superscript number. For example, on page 2, line 71, “reported10”.

Response: We have made changes to the text.